# Learning a Condensed Frame for Memory-Efficient Video Class-Incremental Learning

**Yixuan Pei**[1*]    **Zhiwu Qing**[2*]    **Jun Cen**[3]    **Xiang Wang**[2]    **Shiwei Zhang**[4]
**Yaxiong Wang**[1]    **Mingqian Tang**[4]    **Nong Sang**[2]    **Xueming Qian**[1]

Xi'an Jiaotong University[1]
Huazhong University of Science and Technology[2]
The Hong Kong University of Science and Technology[3]
Alibaba Group[4]
{peiyixuan@stu, wangyx15@stu, qianxm@mail}.xjtu.edu.cn,
{qzw,wxiang,nsang}@hust.edu.cn,
jcenaa@connect.ust.hk, {zhangjin.zsw, mingqian.tmq}@alibab-inc.com

## Abstract

Recent incremental learning for action recognition usually stores representative videos to mitigate catastrophic forgetting. However, only a few bulky videos can be stored due to the limited memory. To address this problem, we propose *FrameMaker*, a memory-efficient video class-incremental learning approach that learns to produce a condensed frame for each selected video. Specifically, FrameMaker is mainly composed of two crucial components: *Frame Condensing* and *Instance-Specific Prompt*. The former is to reduce the memory cost by preserving only one condensed frame instead of the whole video, while the latter aims to compensate the lost spatio-temporal details in the Frame Condensing stage. By this means, FrameMaker enables a remarkable reduction in memory but keep enough information that can be applied to following incremental tasks. Experimental results on multiple challenging benchmarks, *i.e.*, HMDB51, UCF101 and Something-Something V2, demonstrate that FrameMaker can achieve better performance to recent advanced methods while consuming only 20% memory. Additionally, under the same memory consumption conditions, FrameMaker significantly outperforms existing state-of-the-arts by a convincing margin.

## 1    Introduction

Training video action recognition models with all classes in a single fine-tuning stage have been widely studied [6, 13, 30, 34, 37, 50, 51, 61] in recent years. However, in many realistic scenarios, limited by privacy matters or technologies, the different classes can only be presented in sequence, where the previously trained classes are either unavailable or partially available in limited memory. Naively training models will overfit currently available data and cause performance deterioration on previously seen classes, which is named catastrophic forgetting [39]. Class-incremental learning is a machine learning paradigm to fight this challenge when fine-tuning a single model in a sequence of independent classes.

A branch of class-incremental learning methods [5, 7, 9, 12, 21, 33, 45, 59] has achieved remarkable performances in the image domain by re-training a portion of past examples. Meanwhile, some existing video incremental learning methods [41, 53] have demonstrated that storing more past

---

[*]equal contribution

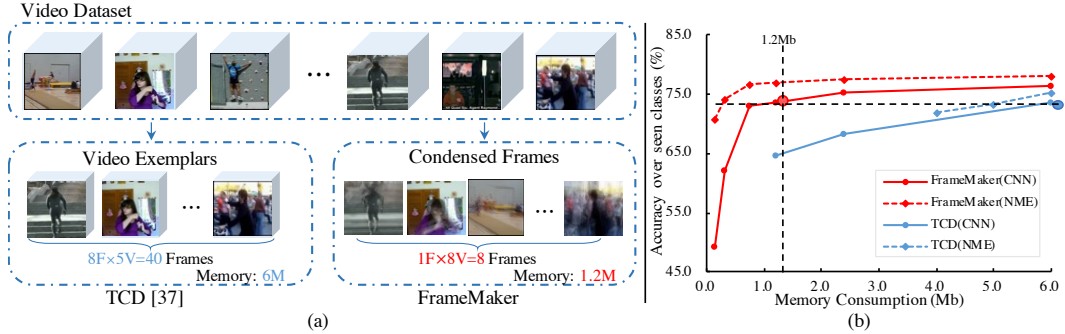

Figure 1: An intuitive comparison between FrameMaker and TCD [41]. (a) Compared with TCD, FrameMaker stores only one condensed frame for each video exemplar. (b) Illustration of memory consumption and accuracy curve. FrameMaker costs 1.2Mb memory, which slightly exceeds the performance of TCD with 6.0Mb memory.

examples can improve the ability to mitigate forgetting. Nevertheless, videos are information-redundant and require a high memory load to be saved, hence it is impractical to store a good deal of training videos for each class. Therefore, herding strategy [45] is adopted to select a small set of representative videos for the exemplar memory [41, 53]. vCLIMB [53] tries to down-sample frames for each video to reduce the memory consumption, and imposes a temporal-consistency regularization for better performance. Despite the remarkable performance, these methods still demand to conserve multiple (*i.e.*, 8-16) frames for each representative video, leading to non-negligible memory overhead, which limits their further real-world applications.

To remedy the above weaknesses, we present FrameMaker, a memory-efficient video class-incremental learning approach. Concretely, the proposed FrameMaker contains two components, Frame Condensing and Instance-Specific Prompt. In Frame Condensing, FrameMaker assigns a learnable weight to each frame, and then these weights are adjusted to make the weighted sum frame *i.e.*, the condensed frame, share the same embedding features with the original video clip. Obviously, the collapsed temporal dimension and mixed spatial cues by Frame Condensing may degrade the accuracy of action recognition. To compensate the missing spatio-temporal details, inspired by Task-Specific Prompt [36], we further propose Instance-Specific Prompt, which learns a set of parameters for each condensed frame to perform fine-grained pixel-wise modifications. In this way, the memory required to store a single video can be remarkably compressed. An intuitive comparison is given in Figure 1(a). On the premise of comparable performance with TCD [41], FrameMaker significantly reduces memory overhead by 80%. Moreover, the comparison curve displayed in Figure 1(b) further indicates that FrameMaker achieves a consistent performance improvement over recent TCD with the same memory consumption.

In summary, we make the following contributions: 1) We present a memory-efficient video class-incremental learning method FrameMaker, which significantly reduces the memory consumption of example videos by integrating multiple frames into one condensed frame; 2) For the first time, we show a novel perspective of Prompt, *i.e.,* Instance-Specific Prompt, and demonstrate that it can embed effective spatio-temporal interactions into learned condensed frames for video class-incremental learning; 3) Experimental results reveal that FrameMaker consistently outperforms recent state-of-the-art methods with the same memory overhead on three challenging benchmarks (HMDB51, UCF101, and Something-Something V2).

## 2 Related Works

### 2.1 Class-Incremental Learning

The class-incremental learning has been well-studied in **image domain** [1, 26, 32, 62], existing approaches could be grouped into knowledge distillation-based methods [20], memory-based methods [40, 45, 46], and architecture-based methods. **In video domain**, class-incremental learning for action recognition is still an underexplored area. The existing solutions are designed for better temporal constraints, such as decomposing the spatio-temporal features [64], exploiting time-channel importance maps [41], and applying temporal consistency regularization [53]. Notably, both [41] and

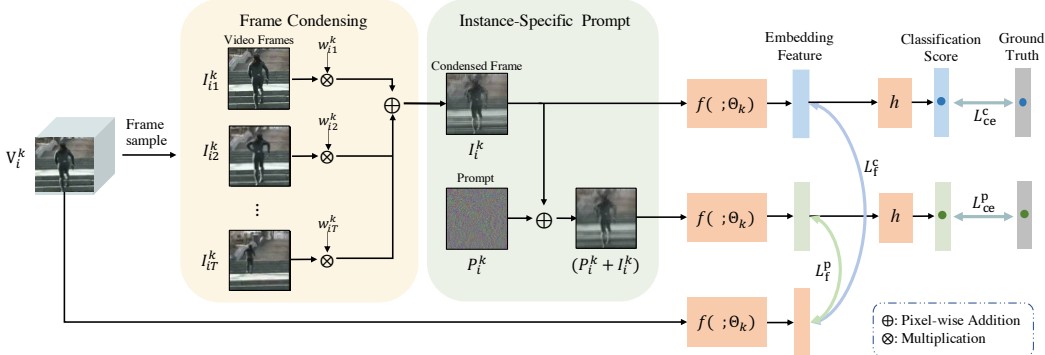

Figure 2: An overview of the proposed FrameMaker, which mainly contains Frame Condensing and Instance-Specific Prompt. For each selected representative video $V_i^k$, we first uniformly sample $T$ frames and learn a condensing weight $w_i^k$ for each frame to perform Frame Condensing. Then a learnable prompt $P_i^k$ is introduced for the condensed frame to compensate the lost spatio-temporal information. Finally, distillation loss and cross entropy loss are both used to guide the optimization of condensing weights and prompt.

[53] have shown that more stored examples in memory can effectively encourage the performance. However, FrameMaker attempts to minimize the stored frames for each video by a pretty maneuver, which can provide decent performance with slight storage requirements.

## 2.2 Prompt-Based Learning

Prompting is an emerging transfer learning technique in natural language processing [36], which adapts the language model to downstream tasks by a fixed function. Since the design of prompting functions is challenging, the learnable prompts [29, 31] are proposed to learn task-specific knowledge parameters for the input. Recently, in the visual domain, some works have also emerged to adapt the pre-trained models to new tasks by adding prompt tokens [24] or pixels [3] in the data space. Besides, DualPrompt [57] and L2P [56] get rid of the rehearsal buffer in continual learning by learning a set of task-specific parameters. In this paper, we draw inspiration from prompting, and present an Instance-Specific Prompt to replenish the collapsed spatio-temporal cues for condensed frames.

## 2.3 Action Recognition

The approaches designed for action recognition task can be grouped into three categories: 2D CNN, 3D CNN and Transformer-based methods. 2D CNN based methods is more efficient [19, 48], while 3D CNN [6, 10, 15, 16, 43, 50–52, 55, 61] achieves better accuracy in action recognition at the cost of more computation cost. With the rapid development of vision transformer [22, 35, 38], researchers also apply it into the video recognition [2, 4, 13, 37], which is proven to be remarkable effective. In this work, we follow the settings in [41, 64], and employ the Temporal Shift Module (TSM) [34] based on 2D CNN for efficient experiments.

The video summarization and efficient action recognition tasks have provided some methods for key-frame selection and generation, which are related to this work. The video summarization [54, 63, 66] task selects the most representative frames for efficient human understanding, which has a different paradigm with action recognition task, hence we will mainly discuss efficient The efficient video classification task aims to learn a strategy to improve classification accuracy with fewer discriminative frames as input. The approaches designed for this task can be grouped into two categories: rule-based and model-based methods. The former [27, 65] select frames via some pre-defined rules such as motion distribution [65] or channel sampling [25], and the later design a submodel to select or generate key frames [14, 17, 23, 44, 58, 60]. In this paper, we propose model-free frame condensing method for video class-incremental learning, which only learns a few condensing weights and prompting parameters for each instance.

# 3 Method

This section presents the general framework of FrameMaker as shown in Figure 2, which mainly consists of two crucial components: Frame Condensing and Instance-Specific Prompt. Firstly, Frame Condensing is designed to integrate multiple frames into one frame for efficient storage. Then we introduce Instance-Specific Prompt to compensate the collapsed temporal information and the mixed spatial knowledge. Finally, the specific class-incremental training details for subsequent tasks using Condensed Frames are given.

## 3.1 Problem formulation

The purpose of video class-incremental learning is to train a model $F(; \Theta) = h(f(; \theta); \xi)$ parameterized by $\Theta$ as the tasks $\{\mathcal{T}^1, \mathcal{T}^2, \cdots, \mathcal{T}^K\}$ arrive in a sequence, where $f(; \theta)$ is the feature extractor and $h(; \xi)$ is the classification head. Each task $\mathcal{T}^K$ has its specific dataset $D^k = \{(d_i^k, y_i^k), y_i^k \in L^k\}$ whose labels belong to a predefined label set $L^k$ and have not appeared in previous tasks. The memory-based methods [41, 45, 53] have been shown to be effective in preventing catastrophic forgetting. Specifically, after the incremental training step $k$, a memory bank $M^k$ will be established for the dataset $D^k$ in this task to store the representative exemplars or features. Exemplars stored in previous datasets $M^{1:k} = M^1 \bigcup M^2 \bigcup \cdots \bigcup M^k$ will be used in subsequent incremental tasks to mitigate forgetting. At the incremental step $k$, the model $F(; \Theta_k)$ is trained from $F(; \Theta_{k-1})$ with $D'^k = D^k \bigcup M^{1:(k-1)}$ and evaluated on all seen classes.

## 3.2 FrameMaker

**Frame Condensing.** We follow the standard protocol of video class-incremental methods [41, 53], which are based on memory-replay strategy and knowledge distillation. Differently, we devotes to reducing the number of stored frames for representative videos to achieve memory-efficient video class-incremental learning. Formally, we first select a subset of video instances $V^k$ by herding strategy [45] from $D^k$ after the incremental step $k$. Given a video $V_i^k \in V^k$, we then sample $T$ frames $\{I_{i1}^k, I_{i1}^k, \cdots, I_{iT}^k\}$ uniformly and integrate them to condense a more representative one. To this end, we define learnable weights $W_i^k = \{w_{i1}^k, w_{i2}^k, \cdots, w_{iT}^k\}$ for the $i$-th video instance $V_i^k$ in exemplar set $V^k$. And the condensed frame can be calculated as follows:

$$I_i^k = \sum_{t=1}^{T} \frac{e^{w_{it}^k}}{\sum_{t=1}^{T} e^{w_{it}^k}} I_{it}^k \in \mathbb{R}^{C \times H \times W} \ , \tag{1}$$

where $C$, $H$ and $W$ are the size of channel, height and width of input frames, respectively. Next, the condensed frame $I_i^k$ is excepted to empower the same or much similar expressive ability as the original video clip. Hence, the embedding features of the condensed frame extracted from current model should be consistent with the features of the original video clip:

$$L_f^c = \| f(I_i^k; \theta_k) - f(V_i^k; \theta_k) \|^2 \ , \tag{2}$$

where $f(; \theta_k)$ is the feature extractor of current model $F(; \Theta_k)$. This consistency regularization can foster the condensing weights to exploit memory-efficient expressions of video clips. To further improve the adaptability of the condensed frames to the current model, we employ cross entropy loss to supervise the classification confidence from the condensed frame:

$$L_{ce}^c = \text{CrossEntropy}(F(I_i^k; \Theta_k), y_i^k) \ . \tag{3}$$

The full objective function for condensing weights $W_i^k$ is given by:

$$L_c = L_f^c + L_{ce}^c \ . \tag{4}$$

With the methods described above, we can learn an effective representation, *i.e.*, the condensed frame for each video based on current model.

**Instance-Specific Prompt.** However, condensing a video into one frame would inevitably lose the the temporal dynamics and complete spatial information of the original video to a certain extent. Therefore, an intriguing perspective of prompt, termed as Instance-Specific Prompt, is proposed to replenish the missing spatio-temporal cues for the condensed frame.

Specifically, we first construct a learnable *prompt* $P_i^k \in \mathbb{R}^{C \times H \times W}$ for each exemplar video $V_i^k$, which shares the same spatial resolution as its original clip. The prompt $P_i^k$ is then pixel-wise summed with the condensed frame $I_i^k$, and learns to pull the embedding feature of the summed frame $(I_i^k + P_i^k)$ and the video clip $V_i^k$ together, which is similar to Eq. 2:

$$L_f^p = || f((I_i^k + P_i^k); \theta_k) - f(V_i^k; \theta_k) ||^2 . \tag{5}$$

Notably, the Eq. 2 for condensing weights is difficult to learn the satisfactory spatio-temporal features due to the collapse of the temporal dimension. Nevertheless, the Eq. 5 can enrich the representations of condensing frames by introducing more flexible learnable parameters in the input space. Besides, the Cross-Entropy loss is also utilized to enhance the semantic perception:

$$L_{ce}^p = \text{CrossEntropy}(F((I_i^k + P_i^k); \Theta_k), y_i^k) . \tag{6}$$

Theoretically, the condensing weights $W_i^k$ and the prompting parameters $P_i^k$ can be jointly updated by $L_p = L_f^p + L_{ce}^p$. Interestingly, we observed in our practice that the flexible prompt $P_i^k$ leads to the condensing weights $W_i^k$ being under optimized. Hence, we give the final training objective function for frame condensing as:

$$L_{fc} = \alpha L_f^c + \beta L_{ce}^c + \gamma L_f^p + \eta L_{ce}^p , \tag{7}$$

where additional $L_f^c$ and $L_{ce}^c$ are added to achieve stronger constraint on $W_i^k$ for its effective training. $\alpha$, $\beta$, $\gamma$ and $\eta$ are the balance weights for each term, and they are empirically set to 1.0 unless otherwise specified.

After the optimization of condensing weights and prompting parameters, the prompt is added directly to the condensed frame and stored together. And the learned prompts will be frozen when the corresponding task is completed. Therefore, there is no extra memory cost for prompts.

### 3.3 Training

It is worth noting that FrameMaker aims to condense the stored frames for representative videos, and other class-incremental learning steps still follow the standard pipeline. Specifically, when training the incremental step $k$, we use the dataset $D'^k = D^k \bigcup M^{k-1}$ to update the model from $F(; \Theta_{k-1})$, where $D^k$ is the dataset of the task $k$ which consists of videos belonging to $L^k$, and $M^{k-1}$ is memory bank which contains condensed frames generated by FrameMaker. The training samples from $D^k$ and $M^{k-1}$ are alternately fed into the current model according to the proportion of their sample number. To further prevent the catastrophic forgetting, we also employ the knowledge distillation method proposed in PODNet [12] to transfer the knowledge from the previous model $F(; \Theta_{k-1})$ to current model $F(; \Theta_k)$. The overall objective function for task $k$ reads:

$$L_{cil} = L_{ce}^d + L_{ce}^m + L_{dist}^m , \tag{8}$$

where $L_{ce}^d$ and $L_{ce}^m$ are the Cross Entropy losses for new task data $D^k$ and condensed frame exemplars $M^{k-1}$, respectively. And $L_{dist}^m$ is the knowledge distillation loss [12] for condensed frame exemplars.

## 4 Experiments

### 4.1 Experimental Setup

**Datasets and evaluation metrics.** The proposed FrameMaker is evaluated on three standard action recognition datasets, UCF101 [49], HMDB51 [28] and Something-Something V2 [18]. HMDB51 dataset consists of 6.8K videos belonging to 51 classes from YouTube or other websites. UCF101 dataset contains 13.3K videos from 101 classes. Something-Something V2 is a crowd-sourced dataset that includes 220K videos from 174 classes.

For UCF101, the model is trained on 51 classes first, and the remaining 50 classes are divided into 5, 10 and 25 tasks. For HMDB51, we train the base model using videos from 26 classes, and the remaining 25 classes are separated into 5 or 25 groups. For Something-Something V2, we first train 84 classes in the initial stage, and generate the groups of 10 and 5 classes.

To evaluate the performance of class-incremental learning, we infer the test videos from all seen categories after each task, and finally, report the average accuracy of all tasks. Following [41], two

Table 1: Comparison with the state-of-the-art approaches over class-incremental action recognition performance on UCF101 and HMDB51. Our FrameMaker achieves the best performance under all experimental settings.

| Num. of Classes | UCF101 | | | | | | HMDB51 | | | |
| | 10 × 5 stages | | 5 × 10 stages | | 2 × 25 stages | | 5 × 5 stages | | 1 × 25 stages | |
| Classifier | CNN | NME | CNN | NME | CNN | NME | CNN | NME | CNN | NME |
|---|---|---|---|---|---|---|---|---|---|---|
| Finetuning | 24.97 | - | 13.45 | - | 5.78 | - | 16.82 | - | 4.83 | - |
| LwFMC [33] | 42.14 | - | 25.59 | - | 11.68 | - | 26.82 | - | 16.49 | - |
| LwM [9] | 43.39 | - | 26.07 | - | 12.08 | - | 26.97 | - | 16.50 | - |
| iCaRL [45] | - | 65.34 | - | 64.51 | - | 58.73 | - | 40.09 | - | 33.77 |
| UCIR [21] | 74.31 | 74.09 | 70.42 | 70.50 | 63.22 | 64.00 | 44.90 | 46.53 | 37.04 | 37.15 |
| PODNet [12] | 73.26 | 74.37 | 71.58 | 73.75 | 70.28 | 71.87 | 44.32 | 48.78 | 38.76 | 46.62 |
| TCD [41] | 74.89 | 77.16 | 73.43 | 75.35 | 72.19 | 74.01 | 45.34 | 50.36 | 40.47 | 46.66 |
| FrameMaker | **78.13** | **78.64** | **76.38** | **78.14** | **75.77** | **77.49** | **47.54** | **51.12** | **42.65** | **47.37** |

Table 2: Comparison with the top approaches over class-incremental action recognition performance on Something-Something V2.

| Num. of Classes | 10 × 9 stages | | 5 × 18 stages | |
| Classifier | CNN | NME | CNN | NME |
|---|---|---|---|---|
| UCIR [21] | 26.84 | 17.98 | 20.69 | 12.57 |
| PODNet [12] | 34.94 | 27.33 | 26.95 | 17.49 |
| TCD [41] | 35.78 | 28.88 | 29.60 | 21.63 |
| FrameMaker | **37.25** | **29.92** | **30.98** | **22.84** |

Table 3: Ablations for Frame Condensing (FC) and Instance-Specific Prompting (ISP) on UCF101 with 10 steps and HMDB51 with 5 steps.

| Frames | FC | ISP | UCF101 | | HMDB51 | |
| | | | CNN | NME | CNN | NME |
|---|---|---|---|---|---|---|
| All | - | - | 72.09 | 75.70 | 43.38 | **47.00** |
| Random | ✗ | ✗ | 68.64 | 73.96 | 39.59 | 43.48 |
| Random | ✗ | ✓ | 70.71 | 75.04 | 39.81 | 43.74 |
| Average | ✗ | ✗ | 70.82 | 75.45 | 41.84 | 45.45 |
| Average | ✗ | ✓ | 71.51 | 76.23 | 42.59 | 46.46 |
| Condensed | ✓ | ✗ | 72.29 | 76.42 | 42.18 | 46.27 |
| Condensed | ✓ | ✓ | **72.93** | **76.64** | **43.39** | 46.88 |

different metrics are reported, *i.e.,* CNN and NME. CNN refers to training a fully-connected layer for extracted features, which is a standard classification protocol. NME is proposed by iCaRL [45], which assigns the labels for test data by comparing the feature embeddings with the mean-of-exemplars.

**Implementation details.** TSM [34] is employed as our backbone, and we follow the data preprocessing procedure of TSM. For UCF101, we train a ResNet-34 TSM for 50 epochs with a batch size 256 from an initial learning rate 0.04. For HMDB51 and Something-Something V2, we train a ResNet-50 TSM for 50 epochs with a batch size of 128 from an initial learning rate of 1e-3 and 0.04, respectively. All used networks are first pre-trained on ImageNet [8] for initialization. These settings are consistent with TCD [41]. We train all models on eight NVIDIA V100 GPUs and use PyTorch [42] for all our experiments.

## 4.2 Comparison with State-of-the-art Results

This section presents the qualitative comparison of our proposed FrameMaker with the existing class incremental learning approaches under multiple challenging settings on three datasets: UCF101 [49], HMDB51 [28] and Something-Something V2 [18]. For a fair comparison, we use the same exemplar memory size per class, model structure and pre-training initialization as the existing methods [41].

Table 1 summarizes the results on UCF101 and HMDB51, which shows that FrameMaker outperforms other methods consistently under different configurations in terms of both CNN and NME scores. The average accuracy of our FrameMaker surpasses TCD by around 3.0% on UCF101 and 2.0% on HMDB51 in terms of CNN, respectively. These results demonstrate that one condensed frame can also be equipped with an effective spatio-temporal representation.

FrameMaker is compared with recent advanced methods on Something-Something V2 in Table 2. FrameMaker sets new state-of-the-art performance under multiple challenging settings. Although FrameMaker only uses one condensed frame for future replay, it still achieves decent performance on the large-scale motion-sensitive dataset. We speculate that these gains mainly come from two aspects: (1) The proposed Instance-Specific Prompt effectively rescues the lost spatio-temporal details, which

Table 4: Ablations for objective functions in Frame Condensing on UCF101 with 10 steps.

| $L_\mathrm{f}^\mathrm{c}$ | $L_\mathrm{ce}^\mathrm{c}$ | CNN | NME |
|---|---|---|---|
| ✗ | ✗ | 70.82 | 75.45 |
| ✓ | ✗ | 71.35 | 75.76 |
| ✗ | ✓ | 71.96 | 76.07 |
| ✓ | ✓ | **72.29** | **76.42** |

Table 5: Ablations for objective functions in Instance-Specific Prompt on UCF101 with 10 steps.

| $L_*^\mathrm{c}$ | $L_\mathrm{f}^\mathrm{p}$ | $L_\mathrm{ce}^\mathrm{p}$ | CNN | NME |
|---|---|---|---|---|
| ✗ | ✓ | ✓ | 71.83 | 75.94 |
| ✓ | ✓ | ✗ | 72.35 | 76.48 |
| ✓ | ✗ | ✓ | 72.58 | 76.57 |
| ✓ | ✓ | ✓ | **72.93** | **76.64** |

Table 6: The number of frames for Frame Condensing on UCF101 with 10 steps.

| $T$ | CNN | NME |
|---|---|---|
| 2 | 71.62 | 75.89 |
| 4 | 71.70 | 76.12 |
| 8 | **72.93** | **76.64** |
| 16 | 72.42 | 76.11 |

Table 7: Ablations for different backbones for FrameMaker on HMDB51 with 5 steps.

| Backbone | Frames | CNN | NME | Mem. |
|---|---|---|---|---|
| TSM | All | 43.38 | **47.00** | 6.00Mb |
| TSM | FC+ISP | **43.39** | 46.88 | 0.75Mb |
| R3D50 | All | 39.85 | **45.64** | 6.00Mb |
| R3D50 | FC+ISP | **39.88** | 45.08 | 0.75Mb |
| ViT | All | **35.34** | 39.46 | 6.00Mb |
| ViT | FC+ISP | 35.25 | **39.58** | 0.75Mb |

Table 8: Ablations for alternative prompting strategies for FrameMaker on HMDB51 with 5 steps.

| Position | Type | CNN | NME | Mem. |
|---|---|---|---|---|
| Feature | T.-Spec. | 41.67 | 45.71 | 77.07Mb |
| Feature | C.-Spec. | 41.72 | 45.93 | 385.35Mb |
| Feature | I.-Spec. | 42.19 | 46.53 | 1926.75Mb |
| Frame | T.-Spec. | 41.16 | 45.34 | 0.75Mb |
| Frame | C.-Spec. | 42.01 | 45.47 | 0.75Mb |
| Frame | I.-Spec. | **43.39** | **46.88** | 0.75Mb |

is also discussed in Section 4.3. (2) Memory-efficient FrameMaker allows us to store more exemplar videos with less memory consumption, which greatly alleviates catastrophic forgetting.

## 4.3 Ablation Study

In this section, we present ablation studies to analyze the properties and the effectiveness of FrameMaker. If not specified, the ablation studies are performed on UCF101 with 10 steps and HMDB51 with 5 steps. For comparison with the case of storing all frames, we select 5 exemplar videos for each class.

**Frame Condensing and Instance-Specific Prompt.** To show the effectiveness of Frame Condensing and Instance-Specific Prompt, we compare them with the following alternative methods to produce stored frames for future replay: (1) *All* Frames, which simply saves the whole video. (2) One *Random* Frame, which randomly selects one frame for each exemplar video. (3) One *Average* Frame, which averages the input frames for each exemplar video. (4) One *Condensed* Frame, which is generated by our Frame Condensing procedure. The results are reported in Table 3. From the table, in terms of CNN, we observe that Instance-Specific Prompt yields an improvement of 2.07% and 0.22% with randomly sampled frames, 0.69% and 0.75% with averaged frames on UCF101 and HMDB51, respectively, which implies that prompts can always replenish the ignored spatio-temporal cues in video frames. Further, Frame Condensing with Instance-Specific Prompt leads to at least on par or better performance than preserving all frames while using just one condensed frame under all settings on both datasets.

**Loss terms.** To preserve the best spatio-temporal information in the condensed frames, the distillation loss $L_\mathrm{f}^*$ and the cross entropy loss $L_\mathrm{ce}^*$ are both applied for Frame Condensing and Instance-Specific Prompt. Table 4 presents the results with Frame Condensing from several different combinations of loss terms. $L_\mathrm{ce}^\mathrm{c}$ provides more explicit semantic supervision, making the accuracy to step further. Similar experiments for Instance-Specific Prompt are shown in Table 5. One noticeable thing is that once introducing prompt, we only need to calculate the final $L_\mathrm{f}^\mathrm{p}$ and $L_\mathrm{ce}^\mathrm{p}$ to realize the joint update of condensing weights $W_i^k$ and prompt $P_i^k$. However, we observe that the learned weights, in this case, tend to be average, which indicates that $W_i^k$ is under-optimized. We speculate that this is due to the strong plasticity of prompt, which leads to the insufficient optimization of $W_i^k$. To this end, we further employ $L_\mathrm{f}^\mathrm{c}$ and $L_\mathrm{ce}^\mathrm{c}$ for $W_i^k$ as the guidance. The improvement in Table 5 and the visualizations in Figure 3 show that $W_i^k$ receives effective learning.

Table 9: Analysis for memory budget on UCF101 with 10 steps. Here '$x$F $\times$ $y$V=$z$Mb' indicates that $x$ sampled frames from $y$ different videos are stored for each class, and $x$ frames are sampled from each video. We assume that the spatial resolution of frames is 224$\times$224, and the total memory consumption is $z$Mbytes.

| Memory Per Class | 8F $\times$ 1V=1.2Mb | | 8F $\times$ 2V=2.4Mb | | 8F $\times$ 5V=6.0Mb | |
|---|---|---|---|---|---|---|
| Classifier | CNN | NME | CNN | NME | CNN | NME |
| iCaRL [45] | - | 58.05 | - | 60.50 | - | 64.51 |
| UCIR [21] | 61.92 | 65.52 | 66.43 | 67.58 | 70.42 | 70.50 |
| PODNet [12] | 63.18 | 70.96 | 65.93 | 72.78 | 71.58 | 73.75 |
| TCD [41] | 64.52 | 71.96 | 68.40 | 73.30 | 73.43 | 75.35 |
| Memory Per Class | 1F $\times$ 1V=0.15Mb | | 1F $\times$ 2V=0.3Mb | | 1F $\times$ 5V=0.75Mb | |
| FrameMaker | 49.37 | 70.78 | 62.06 | 74.18 | 72.93 | 76.64 |
| Memory Per Class | 1F $\times$ 8V=1.2Mb | | 1F $\times$ 16V=2.4Mb | | 1F $\times$ 40V=6.0Mb | |
| FrameMaker | **73.64** | **76.98** | **75.19** | **77.43** | **76.38** | **78.14** |

**The number of frames used for Frame Condensing.** As discussed in Section 3.2, the $T$ frames are uniformly sampled from a representative video for Frame Condensing. In Table 6, we show the effect of $T$ in video class-incremental learning. We observe that the performance increases as $T$ increases for both CNN and NME. However, the performance is saturated when using 16 frames, which is in line with the conclusion of existing methods [41, 53], *i.e.,* storing more frames in a video does not necessarily deliver performance improvement. This phenomenon reveals that FrameMaker can empower one Condensed Frame to absorb plenty of spatio-temporal features from original clips.

**Different backbones.** To evaluate the applicability of proposed FrameMaker, we replace the backbone with 3D CNN-based method R3D50 [16] and video transformer ViT[11], as shown in Table 7. In these experiments, we simply replicate the condensed frames multiple times along the the temporal dimension to meet the 3D models and Transformers. From the results, we can find that our FrameMaker can consistently achieve comparable performance on the three different backbones with only 12% memory cost. These results demonstrate the better generalization of FrameMaker.

**Alternative prompting strategies.** We conduct experiments to explore the types and positions of the prompts in Table 8. *(i)* We compare our Instance-Specific (I.-Spec.) Prompt (ISP) with the Task- (T.-Spec.) and Class-Specific (C.-Spec.) prompts and observe that our ISP can achieve the best performance when operating on both features and frames. Since the actions in videos have great intra-class variance, simply sharing task and class-specific prompts is insufficient to reserve the important spatia-temporal cues for each instance; *(ii) Positions of prompt*: For the feature settings, we add the prompt on the features of the 4 stages of the ResNet. The feature-based prompt is worse than our frame-based prompt, which may be caused by the mismatch between the running model and fixed prompts during incremental procedure. Moreover, additional features require additional storage space because the corresponding prompt cannot be directly added on the changed feature.

**Memory budget comparison.** We now compare the memory budget of FramMaker with existing approaches. To make a direct comparison, we use the definition of the working memory size in terms of stored frames, following [53]. The results are summarized in Table 9. We can see that our FrameMaker only costs 1.2Mb memory, which surpasses TCD [41] using 6.0Mb by 0.21% and 1.63% in terms of CNN and NME. FrameMaker saves 80% of memory space with the comparable performance. Meanwhile, we further increase the stored videos but keep the same memory with existing methods. FrameMaker can still effectively promote performance and shows superior abilities to fight catastrophic forgetting. Our results are about 3% ahead of TCD in both indicators. Besides, we also attempt to select the same number of videos as the existing methods. However, the accuracy of the trained linear classifier, *i.e.,* CNN, is weaker with a small number of video instances. Interestingly, NME evaluated by the exemplar class centre yielded fairly or even better results, implying that the model suffers only slight forgetting. We hypothesize that this is caused by the poor feature diversity that a few condensed frames are provided. Therefore, the classifier tends to overfit fixed sample features, resulting in poor generalization.

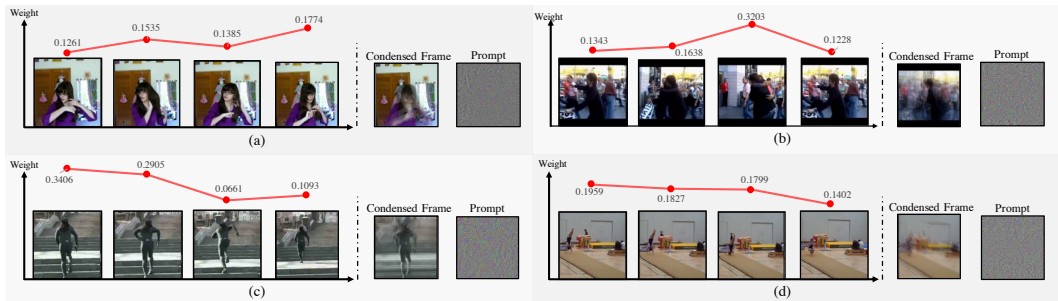

Figure 3: Visualization of Condensing Weights and Instance-Specific Prompts on HMDB51.

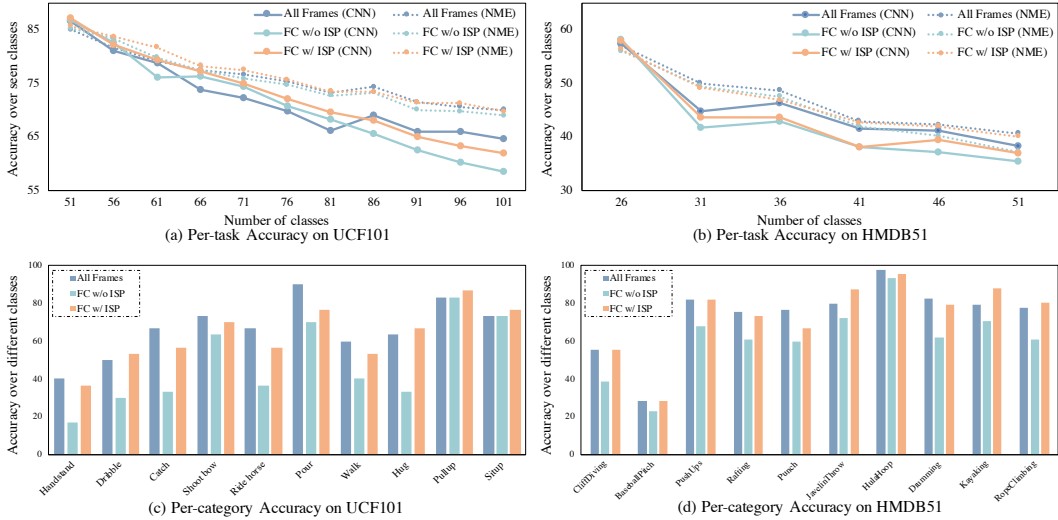

Figure 4: Per-category and per-task accuracy on UCF101 and HMDB51. 'FC' and 'ISP' refer to Frame Condensing and Instance-Specific Prompt, respectively. The plots in (a) and (b) indicate that the introduction of prompt improves accuracy in almost all tasks. The results in (c) and (d) show that our Instance-Specific Prompt significantly improves performance from Frame Condensing in all categories on both datasets.

**Visualization of condensing weights and instance-specific prompts.** To intuitively understand our FrameMaker, we provide visualization of condensing weights, Instance-Specific Prompts, condensed frames and learned prompts in Figure 3. To save space, we uniformly select the most typical 4 frames from $T$ frames for display. Based on this, we have the following fascinating observations: (1) For frames with strong scene bias, the learned weights are more uniform, while those with motion bias are not. For example, the action classification for (a) and (d) in Figure 3 does not depend on a specific frame. Hence, the condensing weight for each frame is roughly the same. In contrast, *stair climbing* and *hugging* in (c) and (b) only occur in some specific frames, and those keyframes are highlighted by condensing weights. This demonstrates that it is reasonable to condense multiple frames into one frame since our proposed Frame Condensing can assist the learning of which frame is meaningful for the action recognition task. (2) It is difficult for humans to absorb useful knowledge from the learned prompts shown in Figure 3, and even they seem to be the same. It should be emphasized that the prompts are different. We try to share the same prompt for all videos to verify this. The experiment is conducted on UCF101 with 10 steps, and the performances decrease by 0.83% and 0.78% in terms of CNN and NME, respectively. This fully validates that our Instance-Specific Prompt is to replenish the specific information for different instances.

**Per-category and per-task analysis.** We depict the task-wise and category-wise accuracy with different forms of stored frames in Figure 4, *i.e.*, "*All Frames*", "*Frame Condensing without Instance-Specific Prompt*" and "*Frame Condensing with Instance-Specific Prompt*". As expected, integrating multiple frames into one condensed frame does undermine performance in almost all cases unless Instance-Specific Prompt is introduced. As shown in Figure 4(a) and (b), the gains on incremental

tasks yielded by Instance-Specific Prompt mainly fall in the later tasks, which implies that the spatio-temporal knowledge absorbed by prompt can effectively alleviate forgetting. Meanwhile, from Figure 4(c) and (d), we also observe that Frame Condensing can lead to the performance degradation of actions with large temporal and spatial variations or short duration, such as "*CliffDiving*", "*Ride horse*" and "*Hug*", etc. Therefore, Frame Condensing with only a few parameters is difficult to capture these complex spatio-temporal interactions. However, Instance-Specific Prompt helps alleviate this issue since it recovers the lost spatio-temporal details in Frame Condensing for each instance.

## 5 Discussions

**Limitations.** Compared with the previous video incremental learning approaches, FrameMaker provides a simple framework that significantly reduces the amount of memory required for each exemplar video. Nevertheless, FrameMaker may fail when the memory budget is highly constrained, which needs to be further explored in future work.

**Societal Impacts.** Although FrameMaker only stores condensed frames in memory buffer for experience replay, which still retains some information about the original video. Therefore, FrameMaker may be used in applications with privacy concerns [47].

**Conclusion.** This paper proposes FrameMaker, a memory-efficient approach for video class-incremental learning. It explores how to learn an effective condensed frame with less memory for each video that can be applied to the future incremental task. FrameMaker mainly consists of two key components, *i.e., Frame Condensing* and *Instance-Specific Prompt*. The former learns a group of condensed weights for each video to integrate multiple frames into one condensed frame, while the latter learns to retrieve the collapsed spatio-temporal structure for the condensed frame. In this way, FrameMaker offers better results with only 20% memory consumption compared to recent advanced methods and sets a new state-of-the-art class-incremental learning performance on multiple challenging datasets. Memory-based class-incremental learning is one of the effective methods to fight forgetting. The efficient utilization of limited space is worth exploring in future work.

**Acknowledgments.** This work is supported by the National Key R&D Program of China under Grant No. 2018AAA0101501, the National Natural Science Foundation of China undergrant 61871435 and 62272380, Fundamental Research Funds for the Central Universities no.2019kfyXKJC024, the Science and Technology Program of Xi'an, China under Grant 21RGZN0017, and by Alibaba Group through Alibaba Innovative Research Program.

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
