# Learning a Condensed Frame for Memory-Efficient Video Class-Incremental Learning
## *Supplementary Materials*

**Yixuan Pei**[1][*]   **Zhiwu Qing**[2][*]   **Jun Cen**[3]   **Xiang Wang**[2]   **Shiwei Zhang**[4]
**Yaxiong Wang**[1]   **Mingqian Tang**[4]   **Nong Sang**[2]   **Xueming Qian**[1]

Xi'an Jiaotong University[1]
Huazhong University of Science and Technology[2]
The Hong Kong University of Science and Technology[3]
Alibaba Group[4]
{peiyixuan, wangyx15}@stu.xjtu.edu.cn,
qianxm@mail.xjtu.edu.cn,
{qzw,wxiang,nsang}@hust.edu.cn,
jcenaa@connect.ust.hk, {zhangjin.zsw, mingqian.tmq}@alibab-inc.com

## A   Additional Experiments for Different Memory Budgets

This is supplementary to Section 4.2 "Comparison with State-of-the-art Results". Limited by space, we only summarize the results of our approach using the same exemplar memory size per class as others and the experiment results of "Memory budget comparison" on UCF101 with 10 steps in the main body. Table A1 supplements the average accuracy of FrameMaker with different memory budgets under all benchmarks of UCF101 and HMDB51 datasets, which is compared with the best performance of TCD [5].

Table A1: Comparison with the state-of-the-art approaches over class-incremental action recognition performance on UCF101 and HMDB51 with different memory budgets.

| | UCF101 | | | | | | HMDB51 | | | |
|---|---|---|---|---|---|---|---|---|---|---|
| Num. of Classes | $10 \times 5$ stages | | $5 \times 10$ stages | | $2 \times 25$ stages | | $5 \times 5$ stages | | $1 \times 25$ stages | |
| Memory Per Class | CNN | NME | CNN | NME | CNN | NME | CNN | NME | CNN | NME |
| $8F \times 5V$=6.0Mb [5] | 74.89 | 77.16 | 73.43 | 75.35 | 72.19 | 74.01 | 45.34 | 50.36 | 40.47 | 46.66 |
| $1F \times 1V$=0.15Mb | 58.75 | 73.97 | 49.37 | 70.78 | 48.31 | 65.86 | 36.08 | 43.82 | 32.53 | 39.00 |
| $1F \times 2V$=0.3Mb | 67.59 | 76.18 | 62.06 | 74.18 | 55.84 | 72.88 | 39.33 | 46.07 | 34.05 | 43.18 |
| $1F \times 5V$=0.75Mb | 73.05 | 76.70 | 72.93 | 76.64 | 68.79 | 75.36 | 43.39 | 46.88 | 38.95 | 44.18 |
| $1F \times 8V$=1.2Mb | 76.09 | 77.88 | 73.64 | 76.98 | 73.57 | 76.76 | 46.25 | 49.02 | 40.80 | 46.97 |
| $1F \times 16V$=2.4Mb | 77.24 | 78.06 | 75.19 | 77.43 | 74.92 | 76.92 | 46.97 | 50.42 | 41.96 | 47.03 |
| $1F \times 40V$=6.0Mb | **78.13** | **78.64** | **76.38** | **78.14** | **75.77** | **77.49** | **47.54** | **51.12** | **42.65** | **47.37** |

As can be seen from Table A1, under other settings, the results of our method are similar to that on UCF101 with 10 steps. When using about 20% of the storage space, our method can achieve better performance than TCD, and the accuracy is further improved when the number of saved exemplars is increased, which also proves the robustness of our approach.

---

[*]equal contribution

36th Conference on Neural Information Processing Systems (NeurIPS 2022).

## B  Additional Analysis of the Balance Weights

This is supplementary to Section 4.3 "Ablation Study". We further discuss the sensitivity to the balance weight for each term in $L_{fc}$ on UCF101 with 10 steps.

We first discuss the sensitivity of $\alpha$ and $\beta$, the balance weights for $L_f^c$ and $L_{ce}^c$ which are used to achieve stronger constraint on $W_i^k$ for its effective training. Table A2 shows the performance of frame condensing (without instance-specific prompt) under various combinations of $\alpha$ and $\beta$. We find the performance under the combination of $\{\alpha = 1, \beta = 1\}$ consistently exceeds others. Although a slightly higher performance is obtained on NME under $\{\alpha = 2, \beta = 1\}$, it is difficult to achieve the same gain on the two different metrics after continuous adjustment.

Table A2: Sensitivity of the performance of Frame Condensing to $\alpha$ and $\beta$ on UCF101 with 10 steps. Default settings are marked in  gray .

| $\alpha(L_f^c)$ | 1 | 2 | 5 | 0.5 | 0.1 | 0.01 | 1 | 1 | 1 | 1 | 1 |
| $\beta(L_{ce}^c)$ | 1 | 1 | 1 | 1 | 1 | 1 | 2 | 5 | 0.5 | 0.1 | 0.01 |
|---|---|---|---|---|---|---|---|---|---|---|---|
| CNN | **72.29** | 72.27 | 71.83 | 72.22 | 72.07 | 71.94 | 71.93 | 71.44 | 72.14 | 71.96 | 71.92 |
| NME | 76.42 | **76.43** | 76.36 | 76.25 | 76.21 | 76.13 | 76.37 | 76.23 | 76.22 | 75.91 | 75.84 |

Then we further analyze sensitivity of $\gamma$ and $\eta$, the balance weights for $L_f^p$ and $L_{ce}^p$, when setting $\alpha$ and $\beta$ as the optimal value, 1. Table A3 shows the performance of FrameMaker (frame condensing with instance-specific prompt) under various combinations of $\gamma$ and $\eta$.

Table A3: Sensitivity of the performance of FrameMaker to $\gamma$ and $\eta$ ($\alpha = 1, \beta = 1$) on UCF101 with 10 steps. Default settings are marked in  gray .

| $\gamma(L_f^p)$ | 1 | 2 | 5 | 0.5 | 0.1 | 0.01 | 1 | 1 | 1 | 1 | 1 |
| $\eta(L_{ce}^p)$ | 1 | 1 | 1 | 1 | 1 | 1 | 2 | 5 | 0.5 | 0.1 | 0.01 |
|---|---|---|---|---|---|---|---|---|---|---|---|
| CNN | 72.93 | **72.94** | 72.92 | 72.88 | 72.64 | 72.67 | 72.90 | 72.85 | 72.92 | 72.80 | 72.73 |
| NME | 76.64 | 76.62 | 76.58 | 76.61 | 76.59 | 76.61 | **76.65** | 76.58 | 76.61 | 76.55 | 76.51 |

We find the performance under the combination of $\{\gamma = 1, \eta = 1\}$ always exceeds others. Some groups of weights make one of the test metrics higher but only $\{\gamma = 1, \eta = 1\}$ obtain the best balance of them. Therefore, we set all the balance weights as 1 finally in our experiments.

## C  Implementation Details

This section provides some additional details about the experiments in the main body.

**Training Details.** This is supplementary to Section 3.3 "Training". We further supplement the elaborate training process and knowledge distillation loss in this section. Figure A1 shows the overall framework of training process.

In the training process of the incremental step $k$, we use the dataset $D'^k = D^k \bigcup M^{k-1}$ to update the model from $F(; \Theta_{k-1})$, where $D^k$ is the dataset of the task $k$ which consists of videos belonging to current classes, and $M^{k-1}$ is memory bank which contains condensed frames generated by FrameMaker after the step $k - 1$. The model classifies the samples from the two datasets respectively and use exemplars in old classes for knowledge distillation.

For the classification task, we adopt cross entropy loss to supervise. $L_{ce}^d$ and $L_{ce}^m$ are the cross entropy losses for video example $V_i^k$ in $D^k$ and condensed frame exemplar $I_i^{k-1}$ in $M^{k-1}$.

$$L_{ce}^d = \text{CrossEntropy}(F(V_i^k; \Theta_k), y_i^k) \,, \tag{A1}$$

$$L_{ce}^m = \text{CrossEntropy}(F(I_i^{k-1}; \Theta_k), y_i^{k-1}) \,, \tag{A2}$$

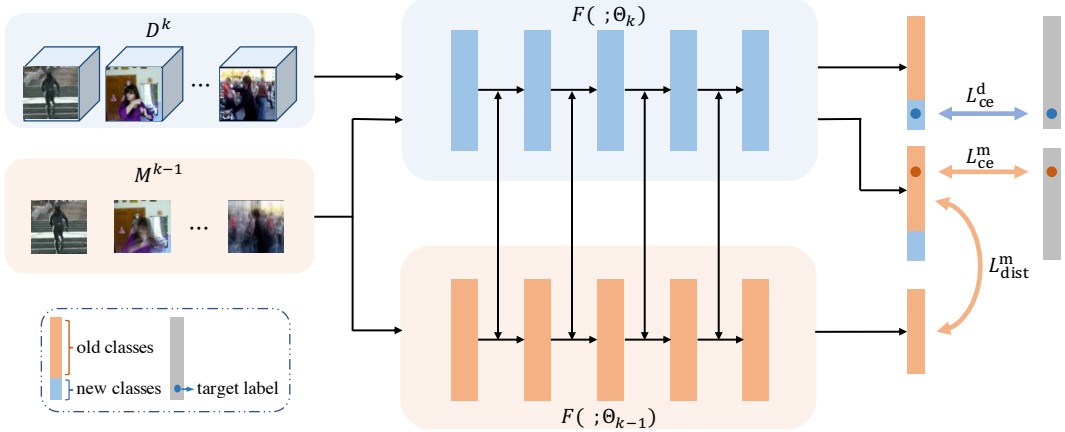

Figure A1: An overview of the training process

where $y_i^k$ and $y_i^{k-1}$ is the ground truth label of $V_i^k$ and $I_i^{k-1}$, respectively.

In order to save the information of old classes better, the knowledge distillation function of PODNet [1] is employed on the old classes data $M^{k-1}$ and it consists of two components, the spatial distillation function $L_{\text{spatial}}$ and the final embedding distillation function $L_{\text{flat}}$.

$$
\begin{aligned}
L_{\text{spatial}}&(f_l(I_i^{k-1}; \Theta_k), f_l(I_i^{k-1}; \Theta_{k-1})) \\
&= \sum_{c=1}^{C} \sum_{h=1}^{H} || \sum_{w=1}^{W} f_{l,c,w,h}(I_i^{k-1}; \Theta_k) - \sum_{w=1}^{W} f_{l,c,w,h}(I_i^{k-1}; \Theta_{k-1}) ||^2 \\
&+ \sum_{c=1}^{C} \sum_{w=1}^{W} || \sum_{h=1}^{H} f_{l,c,w,h}(I_i^{k-1}; \Theta_k) - \sum_{h=1}^{H} f_{l,c,w,h}(I_i^{k-1}; \Theta_{k-1}) ||^2 ,
\end{aligned}
\tag{A3}
$$

where $f_l(; \Theta_k)$ is the output of intermediate convolution layer $l$ of model $F(; \Theta_k)$. C, H, W are the size of channel, height and width of the output feature of layer $l$.

$$
L_{\text{flat}}(f(I_i^{k-1}; \Theta_k), f(I_i^{k-1}; \Theta_{k-1})) = || f(I_i^{k-1}; \Theta_k) - f(I_i^{k-1}; \Theta_{k-1}) ||^2 ,
\tag{A4}
$$

where $f(; \Theta_k)$ is the feature extractor of model $F(; \Theta_k)$.

The final distillation loss of the condensed frame exemplars is given by:

$$
L_{\text{dist}}^{\text{m}} = L_{\text{spatial}}(f_l(I_i^{k-1}; \Theta_k), f_l(I_i^{k-1}; \Theta_{k-1})) + L_{\text{flat}}(f(I_i^{k-1}; \Theta_k), f(I_i^{k-1}; \Theta_{k-1})) ,
\tag{A5}
$$

To simplify, the hyperparameters used in PODNet [1] final distillation loss is ignored.

The total objective function for task $k$ is given by:

$$
L_{\text{cil}} = L_{\text{ce}}^{\text{d}} + L_{\text{ce}}^{\text{m}} + L_{\text{dist}}^{\text{m}} ,
\tag{A6}
$$

which is the same as Eq. 8 in the manuscript.

**Dataset Details.** Following TCD [5], we use three action recognition datasets, UCF101 [8], HMDB51 [3] and Something-Something V2 [2]. The HMDB51 dataset is a large collection of realistic videos from various sources which can be found in https://serre-lab.clps.brown.edu/resource/hmdb-a-large-human-motion-database/. UCF101 dataset is an extension of UCF50 [7] and consists of 13,320 video clips, which can be found in https://www.crcv.ucf.edu/data/UCF101.php. Something-SomethingV2 dataset is a large-scale motion-sensitive dataset that shows humans performing pre-defined basic actions with everyday objects, which can be found in https://developer.qualcomm.com/software/ai-datasets/something-something. For UCF101, we have three settings, $51 + 10 \times 5$ stages, $51 + 5 \times 10$ stages and $51 + 1 \times 25$ stages. For HMDB51, the number of initial task classes is 26, and the remaining 25 classes are separated into 5 or 25 groups. For Something-Something V2, the benchmarks are $84 + 10 \times 9$ stages and $84 + 5 \times 18$ stages. The random seed we use is also the same as TCD [5].

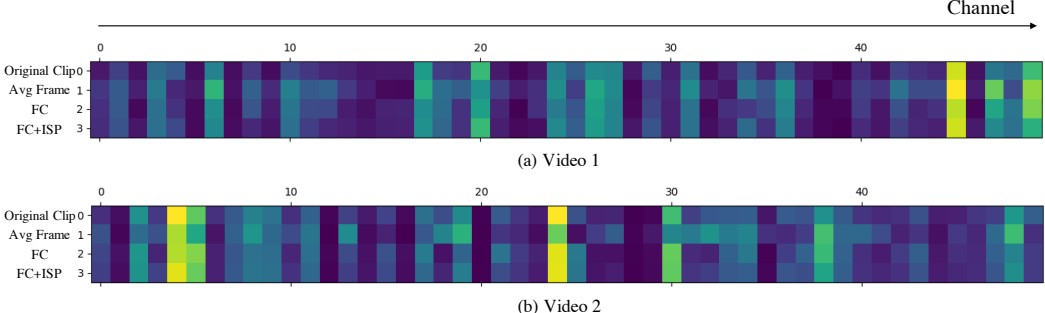

(a) Video 1

(b) Video 2

Figure A2: Channel comparisons between different inputs on HMDB51 . Each line indicates the feature channel activation value obtained with the specified frame as input. FC and ISP are abbreviations of our proposed Frame Condensing and Instance-Specific Prompt, respectively.

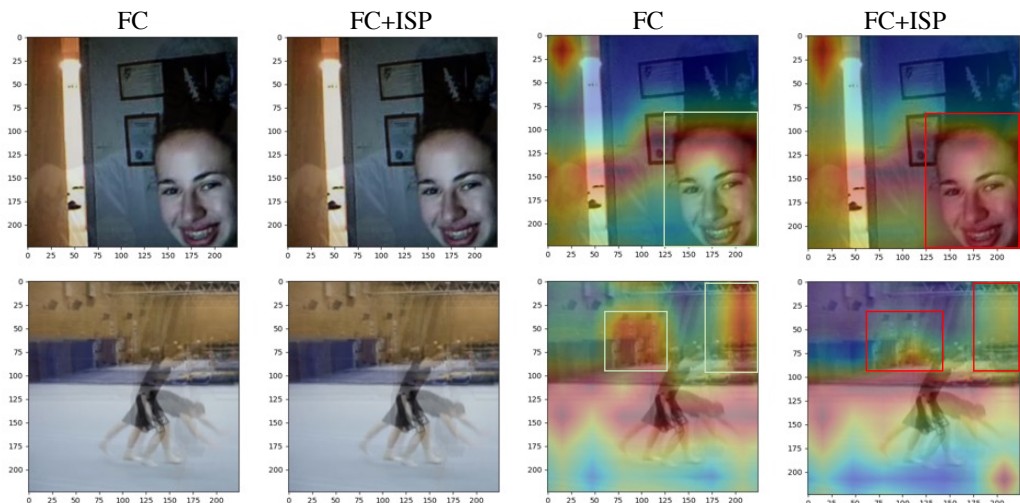

Figure A3: Visualizations of condensed frames and GradCAM maps on HMDB51. FC and ISP refer to proposed Frame Condensing and Instance-Specific Prompt, respectively.

**Model and Hyperparameter Setup.** TSM [4] is employed as our backbone, and we follow the data preprocessing procedure and the basic training setting of TSM. We train a ResNet-34 TSM for UCF101 and a ResNet-50 TSM for HMDB51 and Something-Something V2. Each incremental training procedure takes 50 epochs. We use SGD optimizer with an initial learning rate of 0.04 for UCF101 and Something-Something V2, and 0.002 for HMDB51, which is reduced by half after 25 epochs. For incremental steps, we set the learning rate for all datasets as 0.001 and make the same reduction. The learning rate is set as 0.01 and 0.001 for condensing weights and the instance-specific prompt, respectively, and the total iteration for their optimization is set to be 8k.

**Hardware Information.** We train all models on eight NVIDIA V100 GPUs and use PyTorch [6] for all our experiments.

## D  Visualizations

**Visual comparison of features.** Figure A2 plots visual comparisons of the extracted features with different frames as model inputs. We can summarize that the average frame shows the largest feature difference compared to the features extracted from the original clips. Our proposed Frame Condensing and Instance-Specific Prompt can effectively reduce the feature differences with the original clips and thus effectively improve the feature quality of condensed frames.

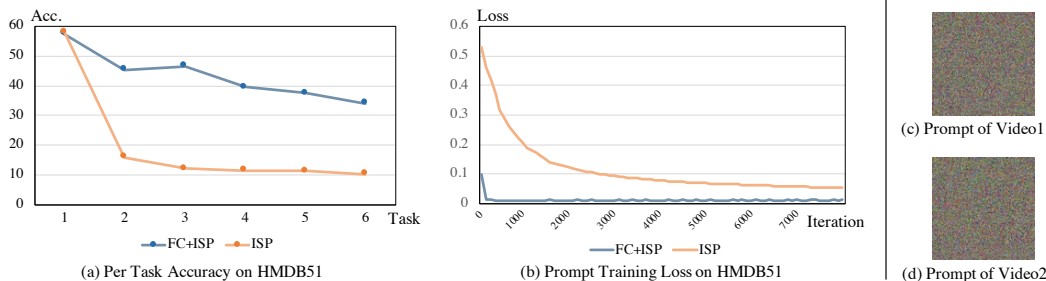

Figure A4: Training prompts without condensed frames. (a): Per-task accuracy on HMDB51. The performance of Instance-Specific Prompt without condensed frames drops quickly. (b): The training loss of prompts on HMDB51. Instance-Specific Prompt without condensed frames is hard to optimize. (c) and (d): Learned prompts of two videos. There is no intuitive semantic information in prompt.

**Visualizations of condensed frames.** Figure A3 plots the condensed frames, the sum of the condensed frames and their prompts, and the activation region generated by the GradCAM technology. From the results, we observe the learned prompts have very small magnitudes (aver- age value is 0.03), hence no significant difference is observed actually. However, from the GradCAM attention maps, we can observe the condensed frames without prompts may focus on background regions, while the learned prompts tend to increase the attention of motion regions or reduce the attention of the backgrounds. We think the reason behind this is that the collapsed temporal dynamics may easily lead the model pay more attention to the background to some extent, while the prompt can guide the model to refocus on the motion region and weaken the interference from backgrounds.

# E   Training Prompt without Condensed Frame

We attempt to train Instance-Specific Prompt with out condensed frames, and the learned prompting parameters are visualized in Figure A4(c) and (d). We observe that the learned prompts have no intuitive semantics. To further understand this phenomenon, we plot the per-task accuracy and the loss curve of the prompts in Figure A4(a) and (b), respectively. We can summarize two points: (i) the learned prompts without condensed frames cannot fight against catastrophic forgetting, which indicates that the semantic information of the learned prompts is weak; (ii) the loss curve of the prompts without condensed frames is much higher than that with condensed frames, which implies that it is difficult to train a single prompts from scratch. Thus, it is crucial that the condensed frames provide a good initialization for the prompts to avoid the local minima of prompts.