# OpenReview forum: "Learning a Condensed Frame for Memory-Efficient Video Class-Incremental Learning"
_NeurIPS.cc/2022/Conference — NeurIPS 2022 Accept_

### Official Review · Reviewer_SSbH · 2022-07-10

**Rating:** 6
**Confidence:** 4
**Soundness:** 3 good
**Presentation:** 3 good
**Contribution:** 2 fair

**Summary:**

This paper presents an approach for video incremental learning with frame condensing and instance prompting. The overall intuition is reasonable, while condensing video into a single frame provides a memory efficiency, it looses the temporal signal, thus instance specific prompting helps to preserves some temporal signal. Experiments are done on HMDB51, UCF101, and Something-Something V2 with good results compared with current approaches. Written presentation is fairly good and enough to understand.

**Questions:**

* Given the proposed architecture are simple and effective, it is natural to ask if it work for the general efficient video classification (considering more on the aspect of memory and compute efficient rather than looking at accuracy). If it works better than TSM, the proposed approach will be much more convincing.

* Are the parameters big/capital \Theta in Eq (3) and Eq (6) shared? what are the intuition? Since Eq (3) applied on only condensed frame while Eq (6) is applied on condensed frame + prompt (expected to be compensated for motions)?

* Experiments are done only on one backbone of TSM, will the method also works on another backbone?

**Limitations:**

* the proposed FrameMaker is currently proved to work for [only] video incremental learning which may limit the impact of the work. If there is experiment to validate the useful of FrameMaker on another task (e.g., efficient video classification not necessary incremental), it will improve the scope of the impact of the work.
* current experiments are done on only one backbone of TSM.

**Strengths And Weaknesses:**

## Strength
* The proposed approach is naturally introduced and the motivations and intuitions for proposing the approach are reasonable and convincing.
* The proposed approach is simple, yet seems to be effective in the problem of incremental learning.
* Experiments show improvements over current methods are consistent across 3 different benchmarks.
* Written presentation is clear and easy to understand.

## Weakness
* The experiments are done using only one backbone architecture, i.e. TSM.
* The reviewer understands the incremental video learning is well-defined and recently studied, but the proposed approach will be more convincing if it works beyond this problem. One closely related problem is efficient video classification using condensed frame + instance-specific prompting?

---

> ### Author Response · Authors · 2022-08-02
> **Response to reviewer SSbH**
>
> Thank you for your acknowledgement of our strengths. We address your concerns as follows:
>
> 1. **Results on more network architectures.**
> Thanks for your careful suggestions. Actually, we apply TSM as the backbone mainly for the fair comparisons with existing video incremental learning methods. Here we use R3D and ViT models to perform an incremental learning task with 5x5 stages on HMDB51 dataset, and the related experimental results are reported in Table R1. In these experiments, we inflate the condensed frame along the temporal dimension to fit the input of the R3D and Transformer models. From the results, we can find that the proposed FrameMaker can achieve comparable performance when comparing with that using all frames as inputs on the three backbones, while the memory consumption drops 87.5\%. These results demonstrate the good generalization of FrameMaker. Moreover, we have added these experimental analysis in the revised manuscript, please refer to Table 7.
>
>     ***Table R1*** Ablations for FrameMaker with different backbones on HMDB51. The incremental learning setting is 5x5 stages.
>     | Backbone | Frames | CNN | NME | Memory Per Class |
>     | :---: | :---: | :---: | :---: | :---: |
>     | TSM | All | 43.38 | **47.00** | 8Fx5V=6.00Mb |
>     | TSM | FC+ISP | **43.39** | 46.88 | 1Fx5V=0.75Mb |
>     | R3D50 | All | 39.85 | **45.64**| 8Fx5V=6.00Mb |
>     | R3D50 | FC+ISP | **39.88** | 45.08 | 1Fx5V=0.75Mb |
>     | ViT | All | **35.34** | 39.46| 8Fx5V=6.00Mb |
>     | ViT | FC+ISP | 35.25 | **39.58** | 1Fx5V=0.75Mb |
>
> 2. **The intuition behind Eq (3) and Eq (6).**
> Thanks for the detailed question. We first clarify that the $\Theta_k$ is shared between Eq (3) and Eq (6). In our method, the CrossEntropy loss function is simultaneously applied to two branches, *i.e.*,  $L^c_{ce}$ for the condensed frames branch ($I_i^k$) and $L^p_{ce}$ for the fused branch ($I_i^k+P_i^k$). Because the unbalance number of the parameters between condensed weights (*i.e*, 8 parameters) and prompt (3xH×W parameters), it tend to be difficulty to optimize the condensed weights using $L^p_{ce}$ alone. Thus we use $L^{c}_{ce}$ to increase the gradient of condensed weights, which can help to improve the quality of the condensed frame.
>
> 3. **FrameMaker for efficient video classification.**
> Thanks for the constructive suggestions. It is an insightful idea to condense multiple video frames into one frame for efficient action recognition. However, recalling that our FrameMaker is designed for the incremental learning task, and the condensed weights and prompt are both instance-specific, it cannot directly used on the efficient action recognition task. To address the problem, we use a lightweight network (*i.e.*, ResNet18) to generate the weights and prompt for each video, thus it can be applied for both training and testing stage. In these experiments, the used models are all initialized by ImageNet pre-trained weights, and we present the results in Table R2. It can be observed that our FrameMaker achieves a comparable performance with saving 44\% computational effort when compared with the baseline, *i.e.*, the original TSM. These are interesting and exciting results, and we will further explore this insightful idea in depth in our future works. Thanks for your valuable suggestions.
>
>
>     ***Table R2*** FrameMaker for efficient action recognition on HMDB51.
>     | Backbone | Approach | Pre-train | Topl | GFLOPs |
>     | :---: | :---: | :---: | :---: | :---: |
>     | RestNet-50 | TSM | ImageNet | 49.54 | 32.7G |
>     | RestNet-50 | FrameMaker | ImageNet | 48.82 | 18.5G |

---

> ### Author Response · Authors · 2022-08-09
> **Message for Reviewer SSbH**
>
> Dear Reviewer SSbH,
> Hope this message finds you well.
> We have updated our manuscript according to your comments, and responded to your questions detailedly. As the discussion period will end in less than one day, we would like to kindly ask whether there is any additional concerns or questions that we might be able to address.
> Thanks very much for your effort!
> Best regards,
> Authors

---

### Official Review · Reviewer_cK7P · 2022-07-11

**Rating:** 6
**Confidence:** 4
**Soundness:** 3 good
**Presentation:** 3 good
**Contribution:** 3 good

**Summary:**

This paper proposes FrameMaker to produce a linearly combined condensed frame and use an instance-specific prompt to represent the lost spatio-temporal details.


**Questions:**

other comments:
Line 98 "Temporal Shit Module" => "Temporal Shift Module"

**Limitations:**

The limitations and societal impact claimed in the paper are reasonable and well-acknowledged.

**Strengths And Weaknesses:**

Strengths:
1. Raw videos have redundant frames for action recognition. Using a condensed frame to represent the whole video is interesting.
2. This paper is well-written and easy to follow.

Weaknesses:
1. Fig.2 $L_f^c$ is hard to optimize, as the condensed frame inherently loses spatio-temporal details. Table 4 shows that $L_f^c$ gives a marginal improvement.
2. Memory comparison is unfair, as FrameMaker stores additional instance-wise prompts. The authors should provide a detailed space analysis for prompts.
3. The compared CIL baselines are however not strong, the authors should consider stronger CIL methods (e.g., AANets[1], DER[2]).

[1] Liu, Yaoyao, Bernt Schiele, and Qianru Sun. "Adaptive aggregation networks for class-incremental learning." Proceedings of the IEEE/CVF Conference on Computer Vision and Pattern Recognition. 2021.
[2] Yan, Shipeng, Jiangwei Xie, and Xuming He. "Der: Dynamically expandable representation for class incremental learning." Proceedings of the IEEE/CVF Conference on Computer Vision and Pattern Recognition. 2021.

---

> ### Author Response · Authors · 2022-08-02
> **Response to reviewer cK7P**
>
> Thank you for your appreciation of our approach. We appreciate your valuable comments, and address your questions as follows:
>
> 1. **$L_f^c$ is hard to optimize because of the lost spatio-temporal details.**
> Thanks for your valuable suggestion. Despite the loss of spatio-temporal information, we actually observe it is not very hard to optimize in our experiments. Recalling that the $L_f^c$ is to force the features extracted from the condensed frames to be same with the original clips, we can encourage the condensed weights to focus on the most informative frames. By this means, we can retain more important information to alleviate catastrophic forgetting. In fact, from the results in Table R7 of **#Reviewer pcyo**, we can find that the similarity between the features of the Condensed Frame and Original Clip is 0.8899, which surpasses the Average Frame by 7.26\% in terms of relative improvement rate. The results show that the $L_f^c$ can in fact reduce the lost spatio-temporal details, which can help us to understand the improvements introduced by $L_f^c$. Beyond that, we can also observe that the Instance-Specific Prompt (ISP) can further reduce the information loss, which indicates the reasonability of our method as well.
>
> 2. **Memory comparisons are unfair.**
> Thanks for your detailed suggestion. Very sorry for that our unclear description confuses you. Actually, the instance-wise prompts do not need additional memory in our method. There are two reasons: **(i)** we directly add the prompting parameters to the condensed frames and store them together; **(ii)** the learned prompts will be frozen when the corresponding task is completed, hence we can fuse them together. Therefore, we no longer need to reserve extra memory for the prompting parameters. A detailed discussion of this question have been supplemented in our revised version (Line 154-156).
>
> 3. **Stronger class incremental learning methods.**
> Thanks for the precious suggestion. We reimplement the image based methods AANets[1] and DER[2] on video incremental learning task and the results are reported in Table R1. For the AANets, we follow the best setting which plugs in the PODNet according the original paper. When implementing DER, we also run with the best setting without using the mask-based pruning strategy. In these experiments, we also the TSM as the backbone and 5x5 stage on HMDB51 dataset for the fair comparison. From the results, we can have the following observations: **(i)** combined with the results in Table 1 in the manuscript, our method can still exceed the SOTA performance with same memory cost, which can prove the effectiveness of our FrameMaker; **(ii)** DER can obtain competitive performance than other methods, and it means that the dynamically expandable structure can work for image-based and video-based task simultaneously. Compared with DER, our FrameMaker is more concise and there is no need to modify the models, and still outperforms it by 1.41\% and 0.77\% in terms of CNN and NME respectively. Therefore, the results further illustrate the effectiveness of our condensed frames on class-incremental task.
>
>     ***Table R1*** Stronger class incremental learning methods. The incremental learning setting is 5x5 stages on HMDB51. The model architecture is TSM.
>     | CIL Methods | CNN | NME | Memory Per Class |
>     | :---: | :---: | :---: | :---: |
>     | AANets [1] | 45.44 | 50.12 | 8Fx5V=6Mb |
>     | DER [2] | 46.13 | 50.35 | 8Fx5V=6Mb |
>     | FrameMaker | **47.54** | **51.12** | 1Fx40V=6Mb |
>
> 1. **There are some grammar errors.**
> Thanks for your careful comments very much. We have carefully correct the grammar errors and typos in our manuscript, and mark them in blue color in the revised version.
>
> [1] Liu Y, Schiele B, Sun Q. Adaptive aggregation networks for class-incremental learning. In CVPR 2021.
>
> [2] Yan S, Xie J, He X. Der: Dynamically expandable representation for class incremental learning. In CVPR 2021.

---

> > ### Comment · Reviewer_cK7P · 2022-08-07
> > **after rebuttal**
> >
> > thanks for the replies. after reading others' comments and feedback, I confirm my rating.

---

> > > ### Author Response · Authors · 2022-08-09
> > > **Thanks for the suggestions of Reviewer cK7P**
> > >
> > > Thank you again for the insightful suggestions that helped improve our manuscript. We appreciate your positive rating.

---

### Official Review · Reviewer_VmcB · 2022-07-11

**Rating:** 7
**Confidence:** 3
**Soundness:** 4 excellent
**Presentation:** 3 good
**Contribution:** 3 good

**Summary:**

This work introduces FrameMaker, a new method of distilling video datasets for class-incremental learning. FrameMaker consists of two components: Frame Condensation and Instance-Specific Prompt. The Frame Condensation learns an optimal weighted average of the exemplar video while the Instance-Specific Prompt learns a per-pixel bias that makes the resulting frame optimal for training a classifier on the respective class.

**Questions:**

Could we see some visualizations of the sums of the condensed frames and the prompts? It would be interesting to see how much this sum image differs from the condensed frame.

I think the authors would also likely be interested in the quickly growing field of “Dataset Distillation.” Several recent works have shown impressive results for this continual learning task in the image domain [A, B].

With these works in mind, I would also like to see some visualizations of an ablation where only the prompt is learned. It seems like the learning of the prompt is very similar to what these dataset distillation methods are doing while the condensed frame acts as a regularizer to ensure the resultant sum image does not overfit to the current sub-task (i.e., subset of classes). I’d like to hear your thoughts on this.

[A] Zhao and Bilen “Dataset Condensation with Distribution Matching” arXiv 2021
[B] Zhou et al. “Dataset Distillation using Neural Feature Regression” arXiv 2022

-----

Edit: raising my rating as authors answered all my initial questions.

**Limitations:**

Yes

**Strengths And Weaknesses:**

Strengths

Based on Tables 1 and 2, FrameMaker out-performs all previous methods on all 3 datasets.

The Figures throughout the paper do a good job explaining the method and its effectiveness.

The authors perform extensive ablation studies to highlight the importance of each component of their method.

Weaknesses

There are a significant number of grammar errors that somewhat distract from the work itself. Rather than list all of them here, I recommend running the text through a grammar-checking program.

Along with Figure 3, it would have been nice to visualize the sum of the condensed frame and the prompt to see just how far the prompt shifts the resulting image away from the weighted sum of frames.

One really tiny thing: the formatting of quotations is inconsistent throughout the paper. Opening quotes should be `` and closing quotes should be ‘’. This will give you the correct-looking curly quotation marks.

---

> ### Author Response · Authors · 2022-08-02
> **Response to reviewer VmcB**
>
>
> Thank you for your thoroughness of our experiments. We appreciate your valuable comments, and address your questions as follows:
>
> 1. **About the grammar errors.**
> Thanks for your careful comments and sorry for our carelessness. We have carefully revised our manuscript again and correct the grammar errors and typos, which marked in blue color.
>
> 2. **Visualizations of the sums of the condensed frames and prompts.**
> Thanks for your constructive suggestions! Figure A3 in our revised supplementary material (the left two columns) illustrates the sum of the condensed frames and the prompts. From the results, we observe the learned prompts have very small magnitudes (average value is 0.03), hence no significant difference is observed actually. However, we try to analyze the impact of prompts with another perspective of the visualization. Specifically, we present the activation region using the GradCAM technology, as shown in Figure A3 (the right two columns). We can observe the condensed frames without prompts may focus on background regions, while the learned prompts tend to increase the attention of motion regions or reduce the attention of the backgrounds (please refer to the heat map in the boxes). We think the reason behind this is that the collapsed temporal dynamics may easily lead the model pay more attention to the background to some extent, while the prompt can guide the model to refocus on the motion region and weaken the interference from backgrounds.
>
> 3. **Discussions about Dataset Distillation.**
> Thanks for the insightful comments. According to your suggestion, we have reviewed the related methods of the dataset distillation area. Here we provide several discussions as follows: **(i)** we conduct the experiments that only the prompt is learned (*i.e.*, ISP-only), and the results are reported in Table R1 and Figure A4 (in the supplementary material). Table R1 shows that FrameMaker can significantly outperform the ISP-only based setting, *i.e.*, 43.39\% vs 19.88\%, which means the retained information of the latter is weak. Take it one step further, from the subgraphs (a) and (b) in Figure A4, we can observe that the ISP-only has a larger loss on both training and testing set, which means it is actually underfitting on the task. Therefore we tend to think the impact of the condensed frame is to implant strong prior and a good initialization for prompt, which can further reduce training difficulty; **(ii)** From the visualization perspective, as shown in subgraphs (c) and (d) in Figure A4, we could find that the learned prompts have no obvious semantics, which may also show the optimization difficulty when training from scratch. However, dataset distillation and incremental learning are a pair of related research topics that fit naturally and deserves our further explore in our future work. Thanks for your suggestion again.
>
>     ***Table R1*** Ablation studies for Instance-Specific Prompt only.
>     | Method | CNN | NME |
>     | :---: | :---: | :---: |
>     | Instance-Specific Prompts only | 19.88 | 23.95 |
>     | Condensed Frames + Instance-Specific Prompts | **43.39** | **46.88** |

---

> > ### Comment · Reviewer_VmcB · 2022-08-07
> > **Response to Response**
> >
> > Thanks so much for addressing all my questions.
> >
> > I'm honestly surprised that the ISP-only results were so bad. DD methods typically work just fine with random noise as initialization, so I'm curious what the problem is here.
> >
> > I'm not expecting any more experiments between now and Tuesday due to it being the weekend and the relatively short notice, but I do have a few curiosities left:
> >
> > 1. I didn't really notice this before, but when comparing $L^c_f$ and $L^p_f$, it seems like the optimal thing would be for $P_i^k$ to just be equal to $0$ and let $I_i^k$ do all the work. Looking at Table 3, I don't think you've done an experiment where both the condensed image and prompt are learned but $L^c_f$ is not used. I'm curious what the results of this experiment would be and if the magnitude of the prompts would be significantly bigger. In the current setup, it almost seems like the two objectives are in conflict with each other and might my suppressing the prompt from growing too large.
> >
> > 2. In the case that it is just an initialization issue, maybe you could try running the ISP-only experiment again except initializing the prompt with the average frame. If it is indeed an initialization issue, maybe this would provide enough of a prior to make the optimization easier.
> >
> > 3. With your current setup, the learned frame weights are a probability simplex. Perhaps it would be beneficial for the weights to be allowed to take a negative value? This could possibly be result in the subtraction of useless background information. Furthermore, in dataset distillation, the optimal distilled images typically have pixel values well outside the range of the training data (e.g., [-40, 40] instead of [-1, 1]). This all being said, I'm curious what would happen if the frame weights were allowed to be freely optimized (i.e., not softmaxed before multiplying by the frames).
> >
> > Again, I'm not holding your rating hostage with these thoughts, I'm just genuinely curious as to what the results would be.
> >
> > Thanks again for your response!

---

> > > ### Author Response · Authors · 2022-08-09
> > > **Response to the Response of Reviewer VmcB (2/2)**
> > >
> > > 2. **More discussions for Instance-Specific Prompts.**
> > > Thanks for your insightful comments. We discuss the ISP by experimental analysis and mathematical analysis as follows. **(i)** Experimental analysis. We perform experiments on HMDB51 with 5 steps, and report the results in Table R3. From the results, we can find that the average frame-initialized ISP can achieve significant improvements than ISP-only by around 20%, which can better show that the condensed frame can indeed provide enough of a prior; **(ii)** Mathematical analysis. Here we will show that initializing the prompts with average frames is mathematically equivalent to summing the prompts and fixed average frames, which further shows that the condensed frames can provide a better initialization. Before that, we try to align the settings between line *(2)* and line *(3)* in Table R3 by separating the random crop operation from the average frame based experiments (*i.e.*, Line 4). For this purpose, we define the average frame as $I_a^k$, the prompt $P_a^k$ which is initialized with zero and the prompt $P_a^{k\prime}$ which is initialized as $I_a^k$. At the beginning of the optimization, $I_a^k+P_a^k = P_a^{k\prime} =\frac{1}{T}\sum_{t=1}^{T}{I_{t}^k}$. During forward propagation, the losses are calculated by:
> > >                 $$L_{I+P}={L(f(I_a^k+P_a^k))}$$
> > >                 $$L_{P}={L(f(P_a^{k\prime}))}$$
> > >     where $f(:)$ is the model and $L(:)$ is the loss function. Since the inputs of $f(:)$ are the same, it is easy to know $L_{I+P} = L_{P}$. During the backward propagation, the gradients are calculated as:
> > >     $$\frac{\partial{L_{\text{I+P}}}}{\partial{P_a^k}} =  \frac{\partial{L_{\text{I+P}}}}{\partial{f(I_a^k+P_a^k)}} \times {\frac{\partial{f(I_a^k+P_a^k)}}{\partial{(I_a^k+P_a^k)}}} \times \frac{\partial{(I_a^k+P_a^k)}}{\partial{(P_a^k)}} = L' \times{f'} \times 1$$
> > >     $$\frac{\partial{L_{\text{P}}}}{\partial{P_a^{k\prime}}} = \frac{\partial{L_{\text{P}}}}{\partial{f(P_a^{k\prime})}} \times {\frac{\partial{f(P_a^{k\prime})}}{\partial{(P_a^{k\prime})}}} = L' \times{f'}$$
> > >     where $I_a^k+P_a^k = P_a^{k\prime}$ and $L_{I+P} = L_{P}$, hence $L'=\frac{\partial{L_{\text{I+P}}}}{\partial{f(I_a^k+P_a^k)}}=\frac{\partial{L_{\text{P}}}}{\partial{f(P_a^{k\prime})}}$, and $f'=\frac{\partial{f(I_a^k+P_a^k)}}{\partial{(I_a^k+P_a^k)}}=\frac{\partial{f(P_a^{k\prime})}}{\partial{(P_a^{k\prime})}}$. Therefore, the parameters $P_a^k$ and $P_a^{k\prime}$ share the same gradient and optimization path, which is also proved by the experimental results (39.83% vs 39.91%). We sincerely thank you for your insightful comments that inspire us to explore the rationale behind our FrameMaker!
> > >
> > >     ***Table R3*** Ablation studies for different Instance-Specific Prompts on HMDB51 with 5 steps.
> > >     | EID | Method | CNN | NME |
> > >     | :---: | :---: | :---: | :---: |
> > >     | 1 | ISP-only by Random Initialization | 19.88 | 23.95 |
> > >     | 2 | ISP-only by Avg. Frame Initialization  | 39.83 | 43.81 |
> > >     | 3 | Avg. Frames without Rand Crop + ISP | 39.91 | 43.80 |
> > >     | 4 | Avg. Frames + ISP | 42.59 | 46.46 |
> > >     | 5 | Condensed Frames + ISP | **43.39** | **46.88** |
> > >
> > > 3. **Softmax function in Frame Condensing.**
> > > Thanks for your valuable suggestion. In Table R4, we train the condensed weights without using the Softmax function for normalization, hence the weights can be freely optimized. From the results, we can find that: **(i)** by analyzing the performance, there is no significant fluctuation, even slightly lower than that with Softmax; **(ii)** by analyzing the statistic of the condensing weights, the results show that the average of weights are very similar (0.145 and 0.125), and the learned weights are rarely negative values. The reason may be that the frames in one video share too many spatial details, hence the negative weights may lead to some meaningful information loss. Therefore, we tend to think the Softmax function is more suitable for our task.
> > >
> > >     ***Table R4*** Ablation studies for Softmax in Frame Condensing on HMDB51 with 5 steps.
> > >     | Softmax | CNN | NME | Max | Min |Mean | Var |
> > >     | :---: | :---: | :---: | :---: | :---: | :---: | :---: |
> > >     | X | 43.35 | 46.81 | 0.6939 | -0.0518 | 0.145 | 9.8e-3 |
> > >     | ✓ | **43.39** | **46.88** | 0.6072 | 0.0126 | 0.125 | 6.7e-3 |

---

> > > ### Author Response · Authors · 2022-08-09
> > > **Response to the Response of Reviewer VmcB (1/2)**
> > >
> > > Many thanks for your positive rating and valuable questions. We address these questions as follows:
> > >
> > > 1. **More discussions for ${L^c_{f}}$ and ${L^c_{p}}$.**
> > > We conduct ablation studies without ${L^c_{f}}$ on UCF101 dataset, and report the results in Table R2. From the results, we can find that the ${L^c_{f}}$ can gain 0.73% when working with ${L^p_{f}}$. We try to analyze the underlying reasons from the two following aspects: **(i)** ${L^c_{f}}$ helps to optimize the condensing weights better. The results show that ${L^c_{f}}$ can improve the variance of the condensing weights, *i.e.*, 'Var. of Weights', from 2.8e-5 to 6.7e-3, which means it increases the difference of the weights for different frames. By this means, we think it could retain more information by selecting important frames. Simultaneously, we also find that the condensed frames are almost equal to the average frames if without ${L^c_{f}}$ in our experiments; **(ii)** Actually, the value of ${L^c_{f}}$  cannot decrease to 0, hence the optimal solution of the prompts also do not equal to 0. We present the average prompts in Table R2, which shows the magnitude can indeed be increased when using ${L^p_{f}}$ alone, from 0.03 to 0.13. To a certain extent, the two objectives conflict with each other.  However, from the final goal of our FrameMaker, they tend to be complementary, which can be proved by the results in Table R2. The performance gain shows that a better condensed frame can be perceived by ${L^c_{f}}$, which is more important to improve the feature representations. We think the reason may be that a better condensed frame can provide more prior for the resultant sum image (${I^k_{i}}+{P^k_{i}}$), and it further reduces the difficulty when optimizing the prompt.
> > >
> > >     ***Table R2*** Ablation studies for ${L^c_{f}}$ and ${L^c_{p}}$ on UCF101 with 10 steps.
> > >     | $L_f^c$ | $L_f^p$ | CNN | NME | Var. of Weights | Avg. Prom. |
> > >     | :---: | :---: |  :---: | :---: | :---: | :---:
> > >     | X | ✓ | 72.20 |  76.35 |   2.8e-5  |  0.13
> > >     | ✓ | X | 72.58 |  76.57 |   7.4e-3   |  0.08
> > >     | ✓ | ✓ | **72.93** | **76.64** | 6.7e-3  | 0.03

---

### Official Review · Reviewer_pcyo · 2022-07-11

**Rating:** 5
**Confidence:** 3
**Soundness:** 2 fair
**Presentation:** 2 fair
**Contribution:** 3 good

**Summary:**

The paper proposes FrameMaker to reduce the memory footprint of video class-incremental learning systems by condensing video examples of past tasks into single frames. Specifically, the authors used a list of learnable parameters to compute weighted average over uniformly sampled frames of each example. This average frame is then combined with a learnable prompt of same dimensionality and fed as input to the main classifier, trained under classification and distillation loss. When compared to prior work on video incremental learning, FrameMaker is shown to improve average accuracies while also reducing the memory needed per class.

**Questions:**

I am looking forward to the authors clarifying the concerns listed in the section above:
- Relation to video summarization/keyframe selection;
- How proposed method works for temporally challenging tasks and a 3D video backbones;
- Alternative prompting strategies;
- Missing baselines and additional evaluation metrics.

Update 8/7: Increased rating post-rebuttal. Most of my concerns and questions have been well addressed in the authors' response.

**Limitations:**

Both limitations and potential societal impacts are addressed in the paper. However, it might help to expand the paragraph to discuss some of the concerns listed above.

**Strengths And Weaknesses:**

**Strengths**

- The proposed solution is well motivated and reduces the memory footprint per class significantly compared to prior methods. It is somewhat surprising that weighted averages of sparsely sampled frames can be used as surrogate for full videos in the memory bank.
- Extensive ablation studies are performed which provides insight into the challenges of video incremental learning problem.

**Weaknesses**

Relation to prior work:
- The problem of frame condensation has high affinity to well-studied areas like video summarization and keyframe detection, yet there is no mention of related work or empirical comparisons. The authors should clarify key differences from these lines of work, in terms of both methodology and ultimate goal.

Concerns about frame condensing method:
- From my understanding, the proposed system requires a video model that can operate on single-frame inputs, such as TSM. For the majority of 3D CNN-based methods or video transformers it is not apparent how to perform forward pass on the condensed frame using the same video model.
- Another suspicion is that averaging the frames can be effective for appearance-biased datasets like UCF101. On Sth v2, many classes are only differentiated through the ordering of frames (move sth from left to right, right to left etc.). It is unlikely that the proposed method can generate informative examples for these temporally challenging scenarios and it might be the reason that improvements over the baseline is less significant.
- Using prompting for incremental learning is a promising direction, but having instance-level prompts of the same dimension as input image seems like an unnatural choice. Have the authors experimented with prompting at the feature level, or sharing prompts across instances of the same class? Also, do the additional parameters introduced by prompting count towards the memory budget?

Concern about experiments:
- Some natural baselines are not compared against, such as using an unweighted average of frames (table 3), or learning task-specific prompts (instead of one prompt per instance). These are necessary to show that the complexities of the proposed system are essential.
- While the visualizations of fig. 3 are nice, I'd imagine there can be more direct metrics to evaluate the quality of frame condensation, such as directly comparing the model output using original clip vs. condensed frame (similar to eq. 2, but evaluated at a later stage of continual learning).

---

> ### Author Response · Authors · 2022-08-02
> **Response to reviewer pcyo (4/4)**
>
> 5. **Missing baselines.**
> Thanks for the careful suggestions. In Table R3, we analyse different methods for frame generation. From the results, we observe that the averaged frames on UCF101 and HMDB51 datasets outperforms the results by using random frames, but is still worse than FrameMaker. Meanwhile, we note that our proposed ISP can consistently improve the performance with averaged frames and random selection, which further demonstrates the effectiveness and generalization of ISP.
>
> 6. **The memory budget of prompting parameters.**
> Thanks for your valuable suggestions. Actually, no additional storage is required because the learned prompting parameters will be directly summed on the condensed frames when the corresponding task is complete. The prompts will no longer be trained when replaying the condensed frames in the later incremental learning tasks. Therefore, our proposed FrameMaker is a memory-friendly solution for video class-incremental learning.
>
> 7. **Additional evaluation metrics.**
> Thanks for the useful suggestion. To better understand the impact of condensed weights and prompting parameters on feature quality, we analyze this at following two levels: **(i) Quantitative analysis**. According to your suggestion, we calculate the average cosine similarity between the features extracted from the generated frames (*i.e.*, the average frame and the condensed frame) and original clip on HMDB51, as shown in Table R7. We can find that our condensed frames can have a larger similarity (0.8899) with origin clip than average frame (0.8297), and the proposed ISP can further improve the similarity to 0.9453. The results show that the proposed Frame Condensing and Instance-Specific Prompt can better reduce the information loss obviously; **(ii) Qualitative analysis**. We visualize the features from different inputs in a channel-by-channel manner in Figure A2 in supplementary material. From this perspective, the learnable condensed weights and the prompting parameters can make the features of a single synthetic frame close to the original clip. These analyses can help to better explain the effectiveness of the proposed Framemaker. Thanks for valuable suggestion again.
>
>     ***Table R7*** Feature similarities between different frames and original clip. The models are TSM and trained on HMDB51 with 5x5 stages.
>     | Objects | FC | ISP | Similarity |
>     | :---: | :---: | :---: | :---: |
>     | Average frame and Original Clip | x | x | 0.8297 |
>     | Condensed frame and Original Clip | ✓ | x | 0.8899 |
>     | Condensed frame and Original Clip | ✓ | ✓ | 0.9453 |
>
>
> [1] Zhi Y, Tong Z, Wang L, et al. Mgsampler: An explainable sampling strategy for video action recognition. In ICCV 2021.
>
> [2] Kim K, Gowda S N, Mac Aodha O, et al. Capturing Temporal Information in a Single Frame: Channel Sampling Strategies for Action Recognition[J]. In AAAI 2022.
>
> [3] Qiu Z, Yao T, Shu Y, et al. Condensing a sequence to one informative frame for video recognition. In ICCV 2021.
>
> [4] Wu Z, Xiong C, Ma C Y, et al. Adaframe: Adaptive frame selection for fast video recognition. In CVPR 2019.

---

> ### Author Response · Authors · 2022-08-02
> **Response to reviewer pcyo (3/4)**
>
> 3. **How proposed method works with other video backbones?**
> Thanks for the carefully questions. In our manuscript, for a fair comparison with previous video incremental learning approaches, we apply the same backbone, *i.e.*, TSM. In Table R5, we present the results by use R3D50 and ViT as the backbones. In these experiments, we simply replicate the condensed frame multiple times along the the temporal dimension to meet the 3D models. From the results, we can find that our FrameMaker can consistently achieve comparable performance on the three backbones with only ~12\%  memory cost. These results demonstrate the better generalization of FrameMaker. We have added these experiments in the revised manuscript, please refer for Table 7.
>
>
>     ***Table R5*** Ablations for FrameMaker with different backbones on HMDB51. The incremental learning setting is 5x5 stages.
>     | Backbone | Frames | CNN | NME | Memory Per Class |
>     | :---: | :---: | :---: | :---: | :---: |
>     | TSM | All | 43.38 | 47.00 | 8Fx5V=6.00Mb |
>     | TSM | FC+ISP | 43.39 | 46.88 | 1Fx5V=0.75Mb |
>     | R3D50 | All | 39.85 | 45.64 | 8Fx5V=6.00Mb |
>     | R3D50 | FC+ISP | 39.88 | 45.08 | 1Fx5V=0.75Mb |
>     | ViT | All | 35.34 | 39.46| 8Fx5V=6.00Mb |
>     | ViT | FC+ISP | 35.25 | 39.58 | 1Fx5V=0.75Mb |
>
> 4. **Alternative prompting strategies.**
> Thanks for the constructive suggestions. We conduct experiments to explore the types and positions of the prompts in Table R6. **(i) Types of prompt**: we compare our instance-specific prompt (ISP) with the task- and class-specific prompts, which respectively share the prompt among same task and class. From the results, we can find that our ISP can achieve the best performance when operating on both features and frames. We think the reason is that the actions in videos have great intra-class variance, so simply using task-specific and class-specific prompts is insufficient reserve the important spatia-temporal information for each instance; **(ii) Positions of prompt**: we also explore the position where embed the prompts, *i.e.*, on features or frames, in Table R6. For the feature settings, we add the prompt on the features of the 4 stages of the ResNet. The results show that frame-based prompt surpasses the feature-based prompt, which we think is caused by the mismatch between the running model and fixed prompts during incremental procedure. Moreover, in terms of storage consumption, additional features require additional storage space because the changed feature cannot be directly added on the corresponding prompt.
>
>     ***Table R6*** Alternative prompting strategies. The models are trained on HMDB51. The incremental learning task is 5x5 stages.
>     | Position | Type | CNN | NME | Memory Per Class |
>     | :---: | :---: | :---: | :---: | :---: |
>     | Feature | Task-Specific | 41.67 | 45.71 | 77.07Mb |
>     | Feature | Class-Specific | 41.72 | 45.93 | 385.35Mb |
>     | Feature | Instance-Specific | 42.19 | 46.53 | 1926.75Mb |
>     | Frame | Task-Specific | 41.16 | 45.34 | 0.75Mb |
>     | Frame | Class-Specific | 42.01 | 45.47 | 0.75Mb |
>     | Frame | Instance-Specific | **43.39** | **46.88** | 0.75Mb |

---

> ### Author Response · Authors · 2022-08-02
> **Response to reviewer pcyo (2/4)**
>
> 2. **How the proposed method works for temporally challenging tasks?**
> Thanks for the valuable question. As discussed in our manuscript, the Instance-Specific Prompt (ISP) is designed to compensate the lost spatio-temporal information, which can implicitly supplement the temporal dynamics. From the results in Table R3, Table R4 and Figure 4(d) (Figure 4(d) is in the manuscript), we can summarize following observations: *(i)* the proposed ISP achieves higher improvement on HMDB51 than UCF101 based on Frame Condensing (FC), which are 1.21\% (43.39\% *vs.* 42.18\%) and 0.64\% (72.93\% *vs.* 72.29\%) respectively. The results can demonstrate that the ISP can reserve important temporal information for the action recognition task to some extent; *(ii)* the ISP can also achieve 1.06\% improvements (*i.e.*, 36.19\% *vs.* 37.25\%) on the temporal challenging SSV2 dataset; *(iii)* the Figure 4(d) in the manuscript shows that ISP can achieve obvious improvements on the some categories, such as "Cliff Diving" and "Baseball Pitch", which possess import temporal clues. Therefore, the proposed FrameMaker can still work for some temporally challenging tasks.
>
>     ***Table R3*** Ablations for Frame Condensing (FC) and Instance-Specific Prompting (ISP) on UCF101 and HMDB51.
>     | Frames | FC | ISP | UCF101-CNN | UCF101-NME | HMDB51-CNN | HMDB51-NME | Memory Per Class  |
>     | :---: | :---: | :---: | :---: | :---: | :---: | :---: | :---: |
>     | All | - | - | 72.09 | 75.70 | 43.38 | **47.00** | 8Fx5V=6.00Mb |
>     | Random | x | x | 68.64 | 73.96 | 39.59 | 43.48 | 1Fx5V=0.75Mb |
>     | Random | x | ✓ | 70.71 | 75.04 | 39.81 | 43.74 | 1Fx5V=0.75Mb |
>     | Average | x | x | 70.82 | 75.45 | 41.84 | 45.45 | 1Fx5V=0.75Mb |
>     | Average | x | ✓ | 71.51 | 76.23 | 42.59 | 46.46 | 1Fx5V=0.75Mb |
>     | Condensed | ✓ | x | 72.29 | 76.42 | 42.18 | 46.27 | 1Fx5V=0.75Mb |
>     | Condensed | ✓ | ✓ | **72.93** | **76.64** | **43.39** | 46.88 | 1Fx5V=0.75Mb |
>
>     ***Table R4*** Ablations for Frame Condensing (FC) and Instance-Specific Prompting (ISP) on SSV2.
>     | Frames | FC | ISP | CNN | NME | Memory Per Class |
>     | :---: | :---: | :---: | :---: | :---: | :---: |
>     | Condensed | ✓ | x | 36.19 | 28.77 | 1Fx40V=6.00Mb |
>     | Condensed | ✓ | ✓ | **37.25** | **29.92** | 1Fx40V=6.00Mb |

---

> ### Author Response · Authors · 2022-08-02
> **Response to reviewer pcyo (1/4)**
>
> 1. **Relation to video summarization, key-frame selection.**
> Many thanks for the insightful comments. To address your comments, here we discuss the differences between FrameMaker and the related works, which mainly relate to efficient video classification (EVC) and video summarization tasks, from the three following aspects:
>     - *Different goal.*
>     The goal is different due to the different tasks. The EVC task aims to learn a strategy to improve classification accuracy with few discriminative frames as input, and the learned strategy is then applied in both training and testing stage. While our purpose is to alleviate the storage burden when fighting catastrophic forgetting during the incremental tasks, and the reserved information helps the model to recall historical information. More importantly, our frame generation strategy is only used in training stage, the model still takes multiple video frames as input during testing. We have surved the approaches in video summarization area, and find its goal is to generate  concise synopsis for efficient human understanding. However, there may exist a gap between human understanding and model understanding, and the corresponding methods have a different paradigm with the first two tasks. Therefore we will mainly discuss the more relevant EVC task next;
>     - *Different methodology.*
>     Existing key-frame selection methods in other areas can mainly divided into two categories: rule-based and model-based approaches. The former select frames via some pre-defined rules, *e.g.*, MGSampler, but they can not self-adapt to different models, which may result in a weak generalization, as shown in Table R1. The later need to carefully design a submodel to select frames, which tend to require a large number of training samples and complex training techniques, such as adversarial training in [3] and reinforcement learning in [4]. However, in the incremental learning task, only a small proportion of examples (*e.g.*, 5 past examples) can be retained to save memory, hence the model-based methods may suffer from overfitting in our task, which can be also proved by the results in Table R1. In comparison, our proposed model-free FrameMaker is very lightweight by only learning a few condensing weights and prompting parameters for each instance, which can simultaneously adapt to the model by end-to-end training. Hence our FrameMaker is an effective and efficient for the video incremental learning task.
>
>         ***Table R1*** Different frame selection methods.
>         | Frame Selection | Learnable | CNN | NME | Memory Per Class |
>         | :---: | :---: | :---: | :---: | :---: |
>         | MGSampler[1] | x | 38.55 | 43.38 | 1Fx5V=0.75Mb |
>         | GrayST [2] | x | 38.21 | 43.61 | 1Fx5V=0.75Mb |
>         | IFS [3] | ✓ | 33.71 | 36.66 | 1Fx5V=0.75Mb |
>         | **FrameMaker** | ✓ | **43.39** | **46.88** | 1Fx5V=0.75Mb |
>         | MGSampler[1] | x | 43.24 | 46.97 | 8Fx5V=6.0Mb |
>         | GrayST[ [2] | x | 40.32 | 44.33 | 8Fx5V=6.0Mb |
>         | **FrameMaker** | ✓ | **47.54** | **51.12** | 1Fx40V=6.0Mb |
>
>     - *Experimental comparisons.*
>     First, we compare with other frame selection methods in the incremental task. For time's sake, we only reimplement two rule-baed methods (*i.e.*, MGSampler and GrayST) and one model-based method (*i.e.*, IFS), as shown in Table R1. From the results, we can find that our method can significantly exceed both the two kinds of methods, *e.g.*, surpasses MGSampler and IFS by 4.84\% and 9.68\% respectively, which indicates that our method is indeed suitable to the video incremental learning task. Second, we also explore our FrameMaker in the EVA task. For this purpose, we use a ResNet18 model to predict condensed weights and prompting parameters by input each example itself. We show the results in Table R2, from which we can observe that our method can reduce the 43.43\% FLOPs of the baseline, but with only 0.72\% performance drop. These results show that our method can have a better generalization on the EVA task. Inspired by these results, we will further explore the idea in this direction. Thanks for such a valuable suggestion again!
>
>         ***Table R2*** FrameMaker efficient action recognition.
>         | Backbone | Approach | Pre-train | Topl | GFLOPs |
>         | :---: | :---: | :---: | :---: | :---: |
>         | RestNet-50 | TSM | ImageNet | 49.54 | 32.7G |
>         | RestNet-50 | FrameMaker | ImageNet | 48.82 | 18.5G |

---

> ### Comment · Reviewer_pcyo · 2022-08-07
> **Thank you**
>
> I would like to thank the authors for their detailed response to each of the concerns I raised. I have updated my rating to borderline accept as most of my initial concerns (relation to prior work, comparison to simple baselines, application to 3D models etc.) have been addressed. At this stage, I am still not fully convinced that the instance prompts can effectively capture the temporal structure of the original videos, though this is not sufficient reason to vote for rejection.

---

> > ### Author Response · Authors · 2022-08-09
> > **Thanks for the adjustment of Reviewer pcyo**
> >
> > We would like to express our deepest gratitude for your insightful and valuable comments, and we also appreciate your adjustment of the final rating. We will further improve our manuscript and explore the rationale behind the Instance-Specific Prompts according to the suggestions from all the reviewers.

---

### Meta-Review · Area_Chair_Fjc8 · 2022-08-25

**Recommendation:** Accept
**Confidence:** Certain

**Metareview:**

The reviewers appreciated that the proposed idea is interesting and is well supported by sufficient empirical evidence. There were some concerns in the initial review and the rebuttal successfully addressed most of them. As a result, two reviewers upgraded their ratings. Overall, this paper tackles an important problem of incremental learning and the proposed approach is efficient (memory-wise) and effective (performance-wise). We are happy to recommend acceptance.

**Award:**

No

---

### Decision · Program_Chairs · 2022-09-14

Accept